

# Rockglaciers on the run – Understanding rockglacier landform evolution and recent changes from numerical flow modeling

Johann Müller[1], Andreas Vieli[1], Isabelle Gärtner-Roer[1]

[1]Department of Geography, University of Zurich, Zurich, 8004, Switzerland

*Correspondence to*: Johann Müller (johann.mueller@geo.uzh.ch)

**Abstract.** Rockglaciers are landforms indicative of permafrost creep and received considerable attention concerning their dynamical and thermal changes. Observed changes in rockglacier motion on seasonal to decadal timescales have been linked to ground temperature variations and related changes in landform geometries interpreted as signs of degradation due to climate warming. Despite the extensive kinematic and thermal monitoring of these creeping permafrost landforms, our

understanding of the controlling factors remains limited and lacks robust quantitative models for rockglacier evolution in relation to their environmental setting.

Here, we use a holistic approach to analyze the current and long-term dynamical development of two rockglaciers in the Swiss Alps. Site-specific sedimentation and ice generation rates are linked with an adapted numerical flow model for rockglaciers that couples the process chain from material deposition to rockglacier flow in order to reproduce observed

rockglacier geometries and their general dynamics. Modelling experiments exploring the impact of variations in rockglacier temperature and sediment/ice supply show that these forcing processes are not sufficient for explaining the currently observed short-term geometrical changes derived from multitemporal digital terrain models at the two different rockglacier. The modelling also shows that rockglacier thickness is dominantly controlled by slope and rheology while the advance rates are mostly constrained by rates of sediment/ice supply. Furthermore, timescales of dynamical adjustment are found to be

strongly linked to creep velocity. Overall, we provide a useful modelling framework for a better understanding of the dynamical response and morphological changes of rockglaciers to changes in external forcing.

## 1 Introduction

Rockglaciers and their dynamics have received much attention in permafrost research and beyond, most prominently by the International Panel on Climate Change (IPCC) in context of impacts of a warming climate on high mountain permafrost

(IPCC, 2014). Time series of rockglacier movement in the European Alps indicate acceleration in permafrost creep in recent decades is related to an increase in ground temperatures (Delaloye et al., 2010b, PERMOS, 2013, Bodin et al., 2015). Furthermore, multitemporal geomorphometric analysis have shown subsidence features and structural disintegration of alpine rockglaciers which are indicative of landform degradation and destabilization (Kääb et al., 2007, Roer et al., 2008b, Bodin et al., 2010, Springman et al., 2013, Micheletti et al., 2015). Many studies have addressed the connection between

mean annual air temperatures (MAAT) and rockglacier dynamics from a descriptive point of view (Ikeda and Matsuoka,



2002a, Roer et al., 2005a, Delaloye et al., 2010b, Springman et al., 2012) or have used modeling approaches to assess rockglacier dynamics (Jansen and Hergarten, 2006, Kääb et al., 2007, Springman et al., 2012). Most of these studies focus on the impact of air or ground temperature on rockglacier creep. Other authors stressed that rockglacier dynamics cannot solely be explained by temperature variations and should integrate flow and controlling environmental factors such as sediment

supply dynamics and landform characteristics (Roer et al., 2005a, French, 2007, Frauenfelder et al., 2008). Rockglaciers have been defined as 'lobate or tongue-shaped bodies of perennially frozen unconsolidated material supersaturated with interstitial ice and ice lenses that move down slope or down valley by creep as a consequence of the deformation of ice contained in them and which are, thus, features of cohesive flow' (Barsch, 1992, p. 176). Such a definition includes information on form, material and process and therefore, the observable rockglacier characteristics are influenced by

sediment and ice input, permafrost conditions and the geomorphological setting which in turn control rheology and landform geometry (Barsch, 1996). Very few studies have addressed rockglacier dynamics in such a holistic approach including backweathering rates, sediment and ice dynamics, and climate variations to gain insight into the long-term evolution of rockglaciers (Olyphant, 1983, Frauenfelder et al., 2008) and were limited regarding validation by observational data.

Recent observations show signs of rockglacier destabilization such as acceleration, subsidence features and structural disintegration (forming of crevasses) at several rockglacier landforms in the Swiss Alps (Kääb et al., 2007, Roer et al., 2008b, Delaloye et al., 2011, Lambiel, 2011, Springman et al., 2013, PERMOS, 2013, Kenner et al., 2014, Bodin et al., 2015). These studies indicate that various factors can lead to such degradation but a common triggering for all the cases has not been identified. These potential factors are most likely connected to the complex combination of the local topography,

the thermal state of the permafrost (climate-induced response) and/or to variations in the sedimentation regime affecting the sediment load during long-term landform evolution.

Numerous remote sensing techniques are available for acquiring data on permafrost creep (see Haeberli et al., 2006 and Kääb, 2008 for an extensive summary), high mountain geomorphometry (Bishop et al., 2003) and high mountain sediment dynamics (Gärtner-Roer, 2012, Heckmann and Schwanghart, 2013, Müller et al., 2014a). This provides the necessary

constraints for a holistic assessment strategy that includes the coupling of relevant landforms and processes and in which sediment supply rates can be quantified, ice volumes estimated and rockglacier rheologies derived (Frauenfelder et al., 2008, Gärtner-Roer and Nyenhuis, 2010).

Here we present such a holistic analysis approach to assess long-term rockglacier evolution and the impacts of variations in temperature, sediment and ice supply on rockglacier geometry and movement. We apply a numerical flow model to two

rockglaciers in the Swiss Alps with different topographic, morphometric and rheological characteristics. The modeling is motivated by observations of topographic and kinematic changes for the two rockglaciers revealing signs of degradation as presented in this study. The aim of the modeling approach is to relate these changes to long-term evolution and short-term adaptation of rockglacier systems to changing environmental factors and ultimately to a better understanding of the currently



observed dominant controls of geomorphological changes. We thereby consider rockglaciers as an integral part of a coarse-debris cascading system in periglacial environments.

## 2 Conceptional approach to high mountain periglacial systems

The topographic evolution of the rockglacier landform relies on the production, transportation and deposition of coarse debris in the periglacial system and the generation and integration of subsurface ice (Wahrhaftig & Cox 1959, Barsch, 1996). The development of rockglaciers is therefore dependent upon the supply of debris from the source headwall(s) and the long-term preservation of an ice matrix or ice core inducing creep (Morris, 1981). Rockglaciers are also dynamic landforms that are influenced by the warming and melting of ice and changes in sediment input. The variations in environmental factors translate into observable changes in geometry and kinematics which can be interpreted as a sign for degradation and/or destabilisation of these permafrost landforms (Roer et al., 2008b, Springman et al., 2013).

Figure 1 shows the theoretical concept of an idealized periglacial mountain slope with a corresponding rockglacier system and builds the conceptual basis for this study. Two main subsystems contribute to the temporal and topographical development of the rockglacier landforms: The upper headwall and talus slope system generate the sediments that are transported into the lower rockglacier system. Besides the sediment input, the rockglacier system is also controlled by the existence, generation and state of subsurface ice and permafrost creep (see Fig. 1). The two subsystems differ on the basis of several characteristics: topographic features, typical landform(s) and the dominating mass transporting processes. Backweathering of the exposed rockwall and resulting rockfall are the most effective mass wasting processes (e.g. Krautblatter et al., 2012, Müller et al., 2014a) and supply the entire system with sediment. Backweathering rates and rockwall dynamics are strongly influenced by the geological structures, lithological conditions as well as by characteristics and dynamics of cleft ice which are in turn thermally controlled (Hasler et al., 2012). The progressive accumulation of sediment and ice on an inclined surface at the foot of the rockwall under permafrost conditions, leads to permafrost creep and develops into a rockglacier (Barsch, 1992, Haeberli et al., 2006). The existence of ice and its properties within the sediment obviously plays an important role as a process agent in these systems and environmental changes influence erosion and transport processes that result in topographical and kinematic changes (White, 1973, Arenson et al., 2004, Haeberli et al., 2006).

→ Insert Figure 1

We transfer this conceptual approach into a numerical flow model that integrates the whole debris process chain and couples the different subsystems and related mass fluxes. It assumes a uniform sediment and ice input from the rockwall to an inclined surface, builds up a talus slope that is supersaturated with ice and then starts to creep as a viscous non-linear media similar to ice. This rheological assumption has repeatedly been used to assess rockglacier kinematics (Wahrhaftig and Cox,



1959, Olyphant, 1983, Whalley and Martin, 1992, Barsch, 1996, Kääb et al., 2007, Frauenfelder et al., 2008). A few studies (e.g. Olyphant (1983), Wagner (1992), Leysinger-Vieli and Gudmundsson (2003) and Frehner et al. (2015)) have demonstrated that such a rheology can in principal be used in a numerical flow model for rockglaciers.

## 3 Recent observations of rockglacier change

The modeling work in this study is motivated by detailed observations of geometric changes of two rockglacier systems, both in Switzerland. We present in this section comprehensive new datasets of the two landforms, rockglacier Huhh1 in the Turtmann valley and the well studied Murtèl rockglacier in the Engadine. Both show changes in surface geometry and kinematic behaviour but have distinctly different landform characteristics (see Table 1 for an overview). In order to assess the controlling mechanisms of rockglacier evolution and potential degradation we use a "backward" approach: We quantify
and discuss distinct observed changes in surface geometry and kinematics of the two rockglaciers, propose potential controlling forcing factors (sediment and ice input as well as ground temperature) and then assess these observations and related forcings with a numerical creep model.

➔   Insert Table 1

### 3.1    Murtèl rockglacier, upper Engadine

The first rockglacier site is the well-studied Murtèl (Hoelzle et al. 2002, Haeberli et al. 2006, Springman et al. 2012) situated below the northern face of Piz Corvatsch (3300 m a.s.l.) in the upper Engadine, in the south-eastern part of Switzerland (UTM N563'112, E5'142'07; zone 32T). The lithology mainly consists of granite and granodiorite. The density of the in-situ
rock types are based on values given in the literature with a density of 2.65-2.75 kg cm$^{-3}$ for granite and 2.7-2.8 kg cm$^{-3}$ for granodiorite (Tarbuck et al. 2011), and studies on backweathering at this site have shown backweathering rates of 2 mma$^{-1}$ (Müller et al., 2014a). Murtèl is one of the best investigated rockglaciers and observations from this permafrost site have been discussed in great detail (see summary in Haeberli et al., 1998). As part of the PERMOS network (Permafrost Monitoring Switzerland), parameters such as borehole temperatures, ground surface temperatures and horizontal velocity
have been monitored since 1987 (Vonder Mühll et al., 2008). The borehole data has revealed a layered internal structure with a shear horizon in 32 m depth where almost all of the deformation takes place (Springman et al., 2012). Attempts to determine the age of this rockglacier (Haeberli et al. 1999, Laustela et al. 2003) obtained an age of 5000 to 6000 a as a minimum value (Haeberli et al., 2003). These values were calculated from present day surface velocity fields assuming constant environmental conditions over the rockglacier development (Kääb und Vollmer 2000, Frauenfelder and Kääb, 2000,
Haeberli et al 2003). The rockglacier is characterized by rather slow creep velocity (0.06-0.13 ma$^{-1}$) and is considered a thick and ice rich landform with a volumetric ice content of 60% (Haeberli et al., 1998, Arenson et al., 2002).



### 3.2    Huhh1 rockglacier, Turtmann Valley, Valais

The second rockglacier is located in one of the hanging valleys of the Turtmann Valley, a tributary of the Rhone valley in southern Switzerland (UTM N401'580, E5'116'471; zone 32T). The valley's lithology mainly consists of Palaeozoic gneisses and schists and based on this lithology rather constant backweathering rates of 2 mma$^{-1}$ are expected (Glade, 2005, Krautblatter et al., 2012, Müller et al., 2014a). The valley stretches from 2400 m a.s.l. to 3278 m a.s.l and is characterized by steep rockwalls, talus cones, a glacier, several moraines of different ages and multiple active and inactive rockglaciers (Roer and Nyenhuis, 2007). The focus within this study lies on the rockglacier Huhh1, which can be considered a thin, moderately fast moving rockglacier (see Tab. 1). This site is also part of the PERMOS network and has undergone several scientific assessments (Rasemann 2003, Roer, 2005, Nyenhuis et al. 2005). There is no direct subsurface information available but Gärtner-Roer (2012) used a semi-quantitative approach to derive the rockglacier thickness and sediment storage assuming an ice volume of 50%-70%. The age of the landform is estimated at 500-600 a using the same approach as Haeberli et al. (2003) and Kääb and Vollmer (2000), where the current velocity fields are assumed to be constant over the rockglacier evolution time. Therefore the age estimates can be seen as minimum ages.

### 3.3    Observations of rockglacier dynamics

Complementary to the PERMOS related kinematic monitoring, we used a combination of remote sensing and terrestrial surveying methods for deriving multitemporal elevation and displacement data in order to assess changes in geometry and creep.

Multitemporal stereophotogrammetric DEMs are available for the analysis between the years 1996 and 2007 for the Murtèl rockglacier. Five high resolutions DEMs have been generated in this study for the Turtmann valley between the years 2001 and 2010; the technical details on these DEMs is given in Tab. 2. New elevation change maps are derived from differencing of the DEMs over the periods 1996 and 2007 for Murtèl and 2001 and 2012 for Huhh1. The limitations concerning processing, uncertainties and application are presented in Kääb and Vollmer (2000), Roer et al. (2005c), Roer and Nyenhuis (2007) and Müller et al. (2014b), and applied in this assessment.

Additionally, kinematic data is available for both rockglaciers from yearly terrestrial geodetic surveys of approximately 20 points as described in PERMOS (2013) and Roer (2005). Horizontal and vertical changes are quantified annually with an accuracy of 1–2 cm. The extracted vertical elevation change from these 3-dimensional displacement vectors is obtained from subtracting the surface-parallel component of the vertical displacement component.

→    Insert Table 2

Based on the above DEMs, new elevation change maps have been derived for both rockglaciers and the subsystem units of the main rockglacier body and contributing talus slope have been identified (Fig. 2). This analysis (over decadal time



periods) showed distinct subsidence features of different magnitudes in the deposition area (outline with green in Fig. 2) of the rockglacier systems. A more detailed assessment of the subsidence is shown in Fig. 3 where the histograms of the yearly subsidence in the deposition area (talus slope/sedimentation area) shows an overall negative average of annual subsidence of -0.04 ma$^{-1}$ for the Murtèl rockglacier and -0.16 ma$^{-1}$ for the Huhh1 rockglacier. Such subsidence features have been

described as signs of permafrost degradation (Roer et al., 2008a, Springman et al., 2013, Bodin et al., 2015) and are assessed by the rockglacier evolution model in Sect. 6.2.4. These observed subsidence rates are calculated considering the uncertainties resulting from the DEM differencing (see Müller et al., 2014b). Additional vertical displacement data from terrestrial surveys conducted from 2001/2009 – 2015 corrected for slope parallel movement agree with the results from the DEM-differencing.

The main lobes of the rockglacier landforms (outlined red in Fig. 2) have also been analysed for subsidence, but the typical 'furrow and ridge structure' of the rockglacier and the topographic dynamics introduced by the creep process complicate the subsidence quantification. Depending on the methodology, annual average surface elevation change in the terminus area of the rockglaciers range from -0.03 ma$^{-1}$ (derived from terrestrial point surveys corrected for slope-parallel movement) to +0.01 ma$^{-1}$ (digital photogrammetry) at Murtèl and -0.2 ma$^{-1}$ (terrestrial point surveys corrected for slope) to +0.06 ma$^{-1}$

(digital photogrammetry) at Huhh1. At the front of both rockglaciers, the elevation change signal is close to the measurement uncertainty and thus, no clear subsidence seems apparent.

Theoretically, subsidence features can result from surface lowering by ice melt (Phillips et al., 2009), reduced ice and sediment input, acceleration of the entire landform (potentially thermally induced, leading to a 'creeping away' and thinning of the rockglacier from its feeding area) or, and most likely, a combination of the above (Roer et al., 2005a).

Our elevation change data also shows the continuing advance of the rockglacier front and the "furrow and ridge" structure. This shows that the rockglacier continues to be active although it is probably no longer fully connected to its sediment source and therefore not in an equilibrium state with the current sedimentation and/or thermal state of the system.

    ➔   Insert Figure 2

    ➔   Insert Figure 3

## 4    Rockglacier evolution- modeling approach

We present here a quantitative rockglacier evolution modeling approach that is based on the conservation of mass and

includes the entire debris process chain in high mountain environments (see Fig. 1).





## 4.1     Geomorphological Setting

In order to initialize and evaluate the numerical model, it is necessary to derive geometric information about the headwall, talus slope and rockglacier. Therefore, the two rockglacier sites have been analyzed according to the concept introduced in Sect. 2 for their along flow geometry and the quantification of sediment input and sediment deposition. Geomorphological

mapping in the field as well as by interpretation of DEMs and orthophotos are used to identify the contributing headwall areas, deposition areas and rockglacier landforms (see Fig. 2). Surface features (e.g. slope, substrate) as well as velocity fields are further used to delimit the different subsystems.

The DEMs served as basis for the geomorphometric analyses to determine spatial dimensions, slope and surface geometry of the periglacial high mountain systems.

## 4.2     Rockglacier creep modeling approach

A 1-dimensional time-dependent numerical flow model is used to simulate the evolution of the rockglacier surface, length and creep velocity along the centre flowline based on a given sediment/ice input and rockglacier rheology. In this study we are not aiming to reproduce the exact evolution or small scale geometric features of the two chosen real-world rockglaciers, but rather use the model to simulate the basic behaviour of a rockglacier body creeping down a slope and investigate the first

order dynamic response of the geometry on changing external factors such as temperature and sediment supply. Specifically, we will investigate potential causes for the observed surface geometry changes (subsidence, front advance and velocity variations) as set out above (Sect. 3.3).

*Rockglacier creep*

For our study we therefore reduce the rheology of the rockglacier to a body of ice-bonded sediment that deforms and creeps like a non-linear viscous material under the influence of gravity, as proposed already in 1959 by Wahrhaftig and Cox (1959) and applied similarly by Olyphant (1983) and Frauenfelder et al. (2008). This rheology can be described by a Glen-type flow law (Glen, 1955) as typically used for glacier ice (Cuffey and Paterson, 2010) which relates the strain rate $\dot{\varepsilon}$ non-linearly to the stress $\tau$

$$\dot{\varepsilon} \propto A\tau^n \tag{1}$$

where $n$ is a flow law exponent that is typically between 2 and 3 for frozen material (Paterson, 2010) and $A$ the rate factor describing the softness of the rockglacier material. Such a constitutive relationship has been applied and discussed in other

studies on rockglacier creep (Olyphant, 1983, Whalley und Martin, 1992, Whalley und Azizi, 1994, Barsch 1996, Azizi und Whalley, 1996, Kääb et al., 2007, Frauenfelder et al., 2008), and is further supported by results from borehole measurements





on real world rockglaciers and shear experiments in the laboratory on rockglacier material (including Murtèl rockglacier; Arenson et al., 2002, Kääb and Weber, 2004, Arenson and Springman, 2005, Frehner et al., 2015).

For simplification we assume the rockglacier material to be a homogenous mixture of ice and sediment, meaning the rheological parameters such as the rate factor $A$ and flow exponent $n$ do not change within the rockglacier body. However,

from boreholes we know that the rheology within rockglaciers is variable (Haeberli et al., 1998) and typically enhanced deformation in ice rich shearing zones are observed, for example, in the case of the Murtèl rockglacier. Such shearing zones are typically near the bottom of the moving body of the rockglacier, where shear stresses are highest, and thus they dominate creep process. Consequently, potential variations in rheology in the material above are not substantially changing the non-linear viscous creep behaviour. The modeled flow is calibrated with observed surface velocities (see Sect. 4.3) and is

dominated by the rheology of the material near the base and thus our modeling implicitly includes the shear zone in its vertically averaged rheology. In addition, the rheology within rockglaciers is generally poorly known and the assumption of a uniform rheology is therefore justified for studying the first order controls of geometric changes.

We further simplify the problem to the case of an infinite sheet of uniform thickness that creeps down an inclined plane (Cuffey und Paterson, 2010, also known as the shallow ice approximation in glaciology (SIA)) and thereby neglect

longitudinal stress gradients. As our focus is the evolution of the surface and not the detailed stress field within the landform, according to Leysinger Vieli and Gudmundsson (2004) this approximation is justified even for relatively high length to thickness ratios such as occurring for rockglaciers. For this 2-dimensional case along a centre flowline the vertical strain rate $\dot{\varepsilon}_{xz}$ is directly related to the shear stress $\tau_{xz}$ through

$$\dot{\varepsilon}_{xz} = A \cdot \tau_{xz}{}^n \qquad (2)$$

where $x$ is the horizontal coordinate along the centre flowline and $z$ the vertical coordinate. The shear stress $\tau_{xz}$ at the base is given by the local surface slope $\frac{\partial s}{\partial x}$ and material thickness $h$

$$\tau_d = \rho_r g \frac{\partial s}{\partial x} h \qquad (3)$$

where $\rho_r$ is the density of the rockglacier material and $\frac{\partial s}{\partial x}$ the surface slope.

Integration of Eq. 2 over the rockglacier thickness results in a surface flow speed $u_s$ from deformation of the rockglacier material of

$$u_s = \frac{2A}{n+1} \left( \rho_r g \frac{\partial s}{\partial x} \right)^n h^{n+1} \qquad (4)$$



and accordingly a vertically averaged horizontal flow speed $\bar{u}$ of

$$\bar{u} = \frac{2A}{n+2} \cdot \left(\rho g \frac{\partial s}{\partial x}\right)^n h^{n+1} \tag{5}.$$

Although this equation is in its form identical to the case of glacier ice (Cuffey and Paterson, 2010), the flow exponent $n$ and the rate factor $A$ (referring to the material softness) are, due to the presence of debris and water within the ice, not necessarily the same. From boreholes and laboratory experiment flow law exponents of rockglaciers have been found between n=1.9 and 4.5 (mean n=2.72) and increase linearly with volumetric ice content $c_{ice}$ of the sediment (Arenson and Springman, 2005). For our relatively high ice contents (60%) a value of n=3 equivalent to the case of ice seems justified and has been used in earlier

studies of Leysinger Vieli and Gudmundsson (2003) and Frauenfelder et al. (2008).

The rate factor $A$ is estimated from observed surface flow speeds by inverting Eq. (3) for $A$ but is known to be influenced by the material temperature. Thus, for the purpose of our temperature forcing experiments and in agreement with known rheological investigations (Paterson and Budd, 1982, Arenson and Springman, 2005) we write the rate factor of the rockglacier material as a product of the temperature dependent part $A^*(T)$ and a scaling factor $f_A$ accounting for the

influence of the debris:

$$A = A^*(T) \cdot f_A \tag{6}$$

For the temperature dependent part we use two approaches. Firstly, as done in Kääb et al. (2007) we use the dependence on

the temperature of pure ice, for which $A^*(T)$ increases exponentially with temperature (see Figure 4; Paterson and Budd, 1982).

Secondly, and probably more realistic for rockglaciers, we follow the description based on shearing experiments of frozen debris material of Arenson (2005) which is given by

$$A^*(T) \propto \frac{2}{T+1} \tag{7}$$

for temperatures between -1 and -4$^{\text{o}}$C. Note that this second version is at warm temperatures above -2$^{\text{o}}$C, as expected for our two cases, more sensitive to temperature warming (Figure 4). For both approaches the temperature dependence is applied at a reference temperature which refers approximately to the real mean annual temperature within the rockglacier body.

    ➔   Insert Figure 4

*Thickness evolution*





The evolution of rockglacier thickness $h$ and rockglacier surface elevation $s$ along the central flowline is calculated from the principle of mass conservation which takes for the 1-dimensional representation the following form (Oerlemans, 2001)

$$\frac{\partial h}{\partial t} = a_r - \frac{1}{w} \cdot \frac{\partial \bar{u} h w}{\partial x}$$

(8)

where $t$ is the time, $a_r$ is the rate of rockglacier material accumulation or removal at the surface (>0 for accumulation; in ma$^{-1}$), $w$ the rockglacier width and $\bar{u}$ the horizontal and vertically averaged flow speed. The geometry of the rockglacier bed transverse to flow is accounted for by assuming a parabolic shaped valley that is prescribed and here assumed to be uniform along the flow.

The evolution of the rockglacier thickness and surface is calculated numerically on a regular grid with 10 m spacing along the centre flowline. Using a standard implicit finite-difference scheme (Oerlemans, 2001) the surface evolution equation (8) is solved at each time step and for all grid points from the depth averaged material flux $\bar{q} = \bar{u} \cdot h \cdot w$ and the material input $a_r$ at the rockglacier surface.

### 4.3    Model input and calibration

*Model geometry*

Approximate bedrock topographies are derived for both rockglaciers from the DEMs and geomorphic mapping (Sect. 4.1) and assumes the bedrock to be roughly parallel to the rockglacier surface. The shapes of the rockglacier beds are approximated to two sections of constant slope that are representative of the two respective rockglaciers. For both rockglaciers we mapped the first 150m of the distance along flow as deposition area and apply there a spatially uniform material accumulation rate at the specified sedimentation rate and ice content whereas further downstream no mass is added or lost at the surface. In the talus slope, where the material is accumulated, we use a slope of 37° (which is slightly below the critical angle of talus slopes) and which is steeper than on the rockglacier part (12° for Murtèl and 27° for Huhh1, see Tab. 4). The respective dimensions and slopes for the two rockglaciers are presented in detail in Tab. 1 and visualized in Fig. 5.

*Material input*

The rockglacier material input rate $a_r$ at the surface is assumed to be positive and uniform on the talus slope and, if not mentioned otherwise, set to zero on the surface of the main rockglacier body. The latter means that in general no sediment or ice is lost at the surface of the main rockglacier. The rockglacier material input at the surface is estimated from the sediment input from the headwall to the talus slope and its respective ice content, which is assumed to be constant in time. The total amount of sediment produced at the headwall is calculated from backweathering rate and headwall area, and is distributed equally over the deposition area (talus slope/accumulation area). Based on in situ measurements (Müller et al., 2014a) and





previous studies (Glade, 2005, Krautblatter, 2012), a backweathering rate of 2 mma$^{-1}$ is used, resulting in an annual sediment input over the entire talus slope of 0.006 m$^3$m$^{-2}$ for Murtel and 0.022 m$^3$m$^{-2}$ for Huhh1. Together with the ice content of the material the accumulation rate of rockglacier material (sediment-ice mixture) is then calculated

$$a_r = \frac{a_s}{(1-c_i)} \tag{9}$$

Based on field studies (Hoelzle et al. 2002) and previous approaches (Gaertner-Roer, 2012) we use an estimated ice content $c_i$ of 60% for both rockglaciers which results in a rockglacier material input rate that is 2.5 times higher than the pure sediment input rate.

*Rockglacier density*

We estimate the density of the rockglacier material $\rho_r$ from the percentage ice content $c_i$ and from the respective densities of ice $\rho_i = 910$ kg m$^{-3}$ and the debris material $\rho_d = 2700$ kg m$^{-3}$

$$\rho_r = (1 - c_i) \cdot \rho_d - c_i \cdot \rho_i \tag{10}$$

*Estimating the rate factor A*

Solving the equation describing surface ice flow from creep of a viscous material (Eq. 4) for the rate factor *A,* we obtain

$$A = A^*(T) \cdot f_A = \frac{(n+1)u_s}{2h^n\tau_d} = \frac{(n+1)u_s}{2h^{n+1}\rho_r g \frac{\partial s}{\partial x}} \tag{11}$$

Using observed surface flow speed data ($u_s$) we can then estimate the corresponding rate factor *A* (respective $f_A$ for a given reference temperature and temperature dependence model) for both rockglaciers. These rate factor values are both substantially lower than the values known for pure ice at similar temperatures (Paterson and Budd, 1982; -1.5$^o$C) which probably reflects enhanced mechanical resistance from the sediment within the ice (Arenson and Springman, 2005).

## 5      Model experiments

The model is applied to the two selected rockglacier systems using the landform specific input parameters in Table 3 and the simplified geometries described in Sect. 4.

➔   Insert Table 3





The build up experiment is documented for Murtèl in Fig. 6 (first 6000 a) and is qualitatively very similar for Huhh1 (shown in Supplementary Fig. S13 to S24). The model starts with an 'empty' topography of bedrock (bedrock topographies in Fig. 5). Initially, it builds up a homogenous sediment-ice body in the talus slope which starts to creep and therefore advances once it reaches a critical thickness and shear stress which occurs roughly after 600 a for Murtèl and 150 a for Huhh1. A

5    rockglacier body is then generated to a characteristic thickness while the front keeps advancing at a roughly constant rate. Further, the growth and geometry change of the rockglacier landform mainly occurs through moving the rockglacier forward at the front. The modeled advance rate is slightly below the surface speed of the main rockglacier body. After a run time of 6000 a for the Murtèl rockglacier and 600 a for Huhh1 which correspond to the ages of the landforms estimated earlier, we obtain geometries (lengths and thicknesses) that are very close to ones currently observed (Fig. 5 and Tab. 4). The actual

10   furrow-ridge structure of the landform cannot be replicated (Fig. 5) due to model design but the overall geometry is well reproduced.

The modeled surface velocities on the main rockglacier lobes range between 0.06 ma$^{-1}$ and 0.09 ma$^{-1}$ for Murtèl and between 0.63 ma$^{-1}$ and 0.79 ma$^{-1}$ at Huhh1 which is in good agreement with the observed values from long term kinematic monitoring (see Tab. 1).

➔   Insert Table 4

Note that the observed rockglacier front shapes and positions slightly differ from the modeled ones as the real bedrock geometries of the rockglaciers are more complex than the assumed uniform mountain slopes.

➔   Insert Figure 5

## 5.2    Perturbation modeling experiments

25   Starting with the rockglacier geometries from the build-up experiments (see Sect. 5.1), we investigate the impact of variations in temperature and material input on rockglacier dynamics. In a first phase we perform two distinctly different perturbation experiments in which we increase the temperature of the rockglacier body by 1°C (Sect. 5.2.1) and in a second independent experiment we completely switch off the material supply to the talus slope (Sect. 5.2.2). In a second step, we then combine these perturbations in temperature and sediment supply.

30   Atmospheric warming is expected to influence both rockglacier temperatures and consequently creep, as well as the production of sediment and incorporation of subsurface ice, but quantification of the latter is highly uncertain (Gruber, 2004, Fischer et al., 2010, Ravanel and Deline, 2011, Schneider et al., 2012). We therefore run varying scenarios for the sediment and ice input with the zero sediment supply being at the extreme end of the spectrum.



For the temperature experiments the chosen step temperature increase of 1°C in the rockglacier depicts a potential warming scenario which roughly refers to a 2°C warming in ground surface temperature (GST) for a fixed position at the permafrost base. The 1°C subsurface warming is also consistent with current and expected future subsurface warming trends based on borehole observations in the Swiss Alps (PERMOS, 2013).

➔   Insert Table 5

Assuming relatively warm reference rockglacier temperatures between -1 and -2°C°, as observed in the European Alps, the Arenson and Springman (2005) temperature dependence gives an increase of the rate factor by a factor of 1.4 to 2.7 (Tab. 5)

for a 1°C warming. The Paterson and Budd (1982) temperature relation however shows almost no dependence on rockglacier temperature and increases only by factor 1.25 with a 1°C temperature increase. If not indicated otherwise, we therefore use the Arenson and Springman (2005) relation in the temperature warming experiments.

The results for the simple temperature and sediment experiments are presented in the following section only for Murtèl rockglacier but the results are qualitatively similar (although of higher absolute magnitude) for the Huhh1 rockglacier. The

more complex and realistic experiments combining variations in temperature and sediment supply are presented for both rockglaciers later in Sect. 5.2.3.

## 5.2.1   Temperature experiment

In a first experiment, a step temperature increase of the entire rockglacier body of 1°C is applied after rockglacier build-up

(at 6000 a), while the sediment supply is held constant. The reference temperature of the rockglacier is set at -1.5°C which results in a rate factor increase by a factor 1.7.

Figure 6 shows the modeled response of the surface geometry, landform thickness and horizontal surface velocity of the Murtèl rockglacier along the central flowline. For reference, the black line in Fig. 6 shows the state of the rockglacier just after build-up (6000 a), immediately before the temperature step change is introduced. The increase in the rate factor causes

an immediate speed-up in horizontal flow of the entire landform by a factor of two roughly (Fig. 6c, yellow lines), which then decays with time (orange to red lines). As a result of the enhanced mass transport, the landform also shows a distinct thinning of up to 0.02 ma$^{-1}$ in the upper part of the rockglacier and on the talus slope (Fig. 6b). At the front the rockglacier continues to thicken and consequently advance, but at accelerated rates as a consequence of enhanced flow speeds (Fig. 6d). With time, both the creep velocity, advance rate and thinning reduce and approach stable values again after about 1000 a for

Murtèl. This state is, apart from the advancing front, stable and characterised by a slightly faster creep velocity and a thinner rockglacier body in order to transport the constant material supply from upstream. Consistent with the creep velocity, the advance rate is also slightly enhanced (Fig. 6d) whereas the volume grows at a constant rate throughout the simulation, reflecting again the constant material supply and mass conservation (Fig. 6e).





Additional model simulations for other reference temperatures of the rockglacier of -1°C and -2°C, show qualitatively very similar results but the absolute rates of change scale proportionally to the enhancement factors in the rate factor given in Tab. 5. The same experiments for the Huhh1 rockglacier show quantitatively similar responses which are of higher absolute magnitude and adjust within 100 a to a new quasi stable state also much faster.

➔ Insert Figure 6

### 5.2.2 Sediment experiment

In a second set of experiments, the influence of variations in sediment and ice supply is investigated by varying the material input $a_r$ but keeping the rockglacier temperature constant. Since there is no empirical data on the impact of temperature increase on sedimentation and ice accumulation rates, a range of changes in material supply rates has been explored.

➔ Insert Figure 7

Figure 7 shows the modeled response for an extreme example in which the ice and sediment input is completely switched-off after rockglacier build-up (at 6000 a). The results show that the rockglacier continues to creep downslope and advance but with reduced velocities that start to decrease from upstream. This slow-down is related to a thinning, reduced slope and driving stresses in the upper part of the rockglacier as the downstream flowing mass is no longer fully replaced by accumulation of material on the talus slope. The rockglacier body essentially creeps downstream without any mass added or removed which is well reflected in the constant volume with time (dotted line in Fig. 7e). The advance rates thereby decrease at relatively low rates. The upper parts of the rockglacier react immediately to the change of material input as this is where the sedimentation is taking place. Note that the maximal thinning rates are only as high as the former material accumulation rate (in case of Murtèl 0.006 ma$^{-1}$, for Huhh1 0.022 ma$^{-1}$).

Experiments with different perturbations in material input rates show qualitatively similar changes but of reduced magnitude (supplement Fig. S1 to S12) and are also evident in differing advance and volume growth rates (Fig. 7d and e).

### 5.2.3 Combined experiment

As atmospheric warming is expected to influence both rockglacier temperatures as well as ice and sediment production, we thus perform a third set of experiments in which we combine the above perturbations.

12 scenarios were run for each rockglacier assuming three different initial thermal states of each rockglacier (see Tab. 5), a




potential warming of 1°C and four different scenarios concerning the material input (see Tab. 6). The corresponding results are shown in the supplementary material.

  ➔   Insert Table 6

In Fig. 8 and 9 we show detailed results for both Murtèl and Huhh1 rockglacier for one representative perturbation experiment in which we used a reference rockglacier temperature of -1.5°C, an increase in temperature of 1°C (corresponding to a rate factor increase by factor 1.7, Tab. 5) and a decrease of material input to 40% of the original value.

10   ➔   Insert Figure 8

Figure 8 and 9 illustrate the evolution of surface geometry and horizontal velocities along the central flow line of Murtèl and Huhh1 rockglacier, respectively. This combined experiment shows an upstream thinning of the "initial" landform in the subsequent years (Fig. 8a and 9a) and a substantial increase in horizontal velocities (Fig. 8b and 9b). The maximum thinning rates occur within the first few decades of the experiment and amount to 1.6 cma$^{-1}$ for Murtèl and 6cma$^{-1}$ for Huhh1 (see Fig. 11). A new stable geometry with advancing front is successively approached, again within roughly 1000 a and 100 a for Murtèl and Huhh1, respectively. The final thickness and velocities of the main rockglacier body are, however, very close to the initial values.

Figure 10 shows the more detailed temporal evolution of geometry and creep velocity at three distinct positions on both rockglaciers. The rockglaciers keep advancing throughout the simulation, with initially slightly enhanced rates caused by the temperature increase and a successive slight slowing down caused by the reduced material accumulation rates. The adjustment times-scales of Murtèl rockglacier are much lower compared to Huhh1, which was also shown by the simple perturbation experiments.

25   ➔   Insert Figure 9

    ➔   Insert Figure 10

The additional combined experiments with a 1°C temperature increase but variable reference temperatures and varying
30   sediment supply rates show qualitatively similar geometric and kinematic responses (see Supplementary Figure S1-S24).

***Subsidence***

The sensitivity of the dynamic response to the initial temperature and to the temperature dependent model has been further analysed for the combined perturbation (1°C warming, 60% reduction in material input) in a sensitivity modeling study (for



detailed results see Supplement). As subsidence is one of the observable quantities from repeated DEM-analysis on real rockglaciers, we summarized the results in terms of maximum thinning rates in Fig. 11.

For the Paterson and Budd (1982) temperature relation, thinning rates are almost independent of the reference rockglacier temperature, but increase with a reduction in material supply and reach maximum thinning rates of 1.8 cma$^{-1}$ and 6.5 cma$^{-1}$

5 for Murtèl and Huhh1, respectively. When using the Arenson and Springman (2005) temperature model, thinning rates strongly increase towards warmer rockglacier reference temperatures, reaching maximum values of 3.4 cma$^{-1}$ and 12.5 cma$^{-1}$ for Murtèl and Huhh1, respectively.

➔   Insert Figure 11

## 6      Discussion

### 6.1      Modeling approach and rockglacier build-up

Based on a continuum approach, our numerical model couples observed sediment input rates and the rockglacier creep process in order to simulate the evolution of creep velocities and surface geometry as well as their dynamic interactions. This

quantitative approach of coupling the relevant subsystems (headwall, deposition area and rockglacier), although highly simplified, was successful in building-up the observed rockglacier geometries and related kinematics (horizontal velocities) within the expected timescales (Tab. 1). The basic dynamic behaviour of a continuously advancing rockglacier body is well reproduced, while the thickness of the main body remains roughly constant.

The modeling of rockglacier build-up shows that besides topographical factors such as slope, the long-term advance rates

and horizontal velocity are predominantly controlled by the rates of material accumulation and rockglacier rheology, whereas the thickness of the main landform seems less sensitive to material supply rates.

The match of observed to modeled velocities and thicknesses should, by model construction, be expected (for similar surface slopes) given that the rate factor A and therefore the viscosity of the rockglacier material has been derived from such observed quantities (Eq. 11), but the agreement supports our modeling approach. More importantly, the material input rates

and build-up times are fully independent estimates and it is therefore not necessarily obvious to get the right rockglacier geometry at the prescribed time.

Our modeled constant advance rates and consistency between modeled and previously estimated rockglacier build-up times further supports the method of back-calculating rockglacier age from current surface velocities (Frauenfelder and Kääb, 2000, Kääb und Vollmer, 2000, Haeberli et al., 2003). Even for the case of temperature perturbations, advance rates of the

30 front do not substantially change in the long-term and thus this 'dating' methods seem still appropriate for alpine rockglaciers. Advance rates are in the long-term however affected by changes in material supply rates. It remains to note that it is actually the vertically averaged velocity, and not the surface velocity, that should match the advance rates. Our modeled



advance rates (derived from an assumed rheology with n=3 in Eq. (5)) corresponds in general to 4/5 of the surface velocity (e.g. modeled surface velocity on the main body of 0.065 ma⁻¹ and an advance rate of 0,054 ma⁻¹ for Murtel rockglacier). Consequently, using surface velocities in back-calculations of rockglacier age may overestimate the age. For many real-world rockglaciers (including Murtèl), rockglacier movement is dominated by deformation in a shear zone near the base and thus surface and vertically averaged creep are almost identical.

Strong simplifications have been made for our modeling approach such as using a homogenous sediment-ice body of uniform temperature and rheology and a spatially uniform and temporally constant material input. The successful rockglacier build-up therefore supports the idea that despite such simplifications rockglacier dynamics and evolution can be reduced to our simple model approach which is based on the historic concept of Wahrhaftig and Cox (1959) and confirms earlier numerical modeling approaches of Olyphant (1983) and Frauenfelder et al. (2008).

The non-linear viscous Glen-type flow-law used here is also supported by laboratory experiments (Arenson and Springman, 2005) and field observations from boreholes (Arenson et al., 2002). However, in reality the involved flow law parameters are, unlike the assumptions of our model, rarely constant in space and time. The flow-law exponent n has been found to increase with ice content (Arenson and Springman, 2005) and relatively thin shear layers with strongly reduced viscosity often dominate rockglacier creep (Hoelzle et al., 2002, Haeberli et al., 2006, Buchli et al., 2013). A more complex ice rheology could in theory and should in the future be included in rockglacier creep models, but currently there is very limited quantitative information available to constrain more complex constitutive relationships. Due to the fact that the creep is also dominated near the base within our model and that we have calibrated our model parameters to observed geometry and velocities, we do not expect the general dynamical behaviour and involved time-scales to be substantially different for other rheological parameters choices. We see our highly reduced approach also as an advantage for identifying the most essential controls and processes in rockglacier evolution.

In our approach the geomorphological mapping of the different subsystems, the quantification of sediment input rates, the ice content and the horizontal velocities are the crucial observational constraints and sufficient to set up a model of rockglacier evolution. However, the simplicity of the model design does not, by construction, allow reproduction of the exact small scale features such as ridges and furrows of the two chosen real-world rockglaciers (Frehner et al., 2015).

## 6.2     Dynamical adjustment to external forcing

The perturbation modeling experiments of applying a sudden change in sediment input, material temperature or a combination of the two provide useful insights into the dynamic and geometric adjustment of rockglaciers to changes in external forcing and therefore also into potential mechanisms that explain observed rockglacier degradation.

The two types of perturbation experiments show similarities but also clear differences in the dynamic response. Both, an increase in the rockglacier temperature as well as a reduction in material input lead to a thinning of the rockglacier and the talus slope whereas the front keeps advancing through thickening. Although the thinning is caused by different mechanisms



(lack of material supply from the rockwall or a runaway of mass through creep acceleration), from an observational point of view the two forcing mechanisms are difficult to distinguish. Importantly, the modeling shows a pronounced thinning occurring in the upper parts of both rockglacier systems (deposition area/talus slope).

The response of horizontal velocities are, for the two types of perturbations, distinctly different. A reduction in material supply results in a decrease of creep velocities from the top, whereas the temperature increase leads to an immediate acceleration of the entire landform, although in the long-term the velocities return to almost pre-perturbation conditions (see Fig. 6c and 7c). The temperature experiments show a brief increase in the advance rates of the rockglaciers, which also return to similar rates as before the perturbation. This occurs in contrast with a continuous reduction of the advance rate at the decreased sediment supply experiment. Therefore, temperature changes within the rockglacier show a strong impact on short term velocity variations whereas changes in the material input determine the long term advance rates and geometry. Similar experiments by Olyphant (1983) and Frauenfelder et al. (2008) focused only on combined experiments and could therefore not address the impact of the individual forcings.

### 6.2.1 Dependence on rockglacier properties

Irrespective of the type of experiments, the absolute magnitude in response (velocity change, thickness change or advance rate) is for the two rockglaciers very different. Relative to the initial quantities however (pre-perturbation velocity, thickness or advance rates), the changes and temporal evolution for both rockglaciers are very similar (Fig. 10). This means that we should expect dynamic changes of rockglaciers to be scalable by their geometric and kinematic characteristics. This scalability is also in good agreement with the observational dataset of multiannual creep variations of rockglaciers in the Swiss Alps (PERMOS, 2013). These show very similar normalized horizontal velocity variations as a potential response to air temperature changes despite their distinctly different characteristics (Delaloye et al., 2010a and PERMOS, 2013). Recent continuous observations of creep velocities on rockglaciers in the Matter Valley in the Swiss Alps confirm this finding even for seasonal timescales (Wirz et al. 2015).

The modeled short-term increase in horizontal velocity for a 1°C warming in rockglacier body is also consistent with the observed speed-up in rockglacier creep of about 300% in the year 2003/2004 with exceptional snow conditions and a very warm summer (Delaloye 2010a).

### 6.2.2 Temperature dependence

Our temperature perturbation experiments assume an immediate warming of the entire rockglacier due to an increase in air temperature and consequently GST. This is, of course, an over-simplification because the vertical heat transfer depends on the energy balance at the surface and heat transport processes and properties within the rockglacier material (Hanson and Hoelzle, 2004). Heat transfer processes should potentially be implemented in future studies (Kääb et al 2007, Scherler et al., 2014). Hence we consider our modeled warming rather as a simple way to investigate the sensitivity to temperature increase.





For relatively thin rockglaciers such as Huhh1, a climatic warming could affect the whole rockglacier thickness in time scales of a few years. For thicker rockglaciers such as Murtèl, it could take several decades for the temperature change to reach the base where most of the deformation actually occurs. This implies that thin, and consequently steep and fast rockglaciers should be more sensitive to warming from the surface.

The sensitivity to temperature warming is also enhanced for relatively warm rockglaciers (with temperatures only a few degrees below freezing), considering the most realistic model for temperature dependence of rockglacier rheology by Arenson and Springman (2005). At -1.0$^{o}$C rockglacier temperature, a warming of 1$^{o}$C results in a 2.7 times increase in the rate factor (and hence creep velocity) whereas the same warming at -2$^{o}$C results only in a 1.4 times higher rate factor. Observed rockglacier temperatures in the Swiss Alps indicate already relatively 'warm' temperatures (close to zero degrees) and show a tendency for further warming (PERMOS, 2013). Such a warming would further amplify the response in acceleration. In contrast, when using a temperature dependence of pure ice (Paterson and Budd, 1982), as done in an earlier attempt of modeling the impact of temperature change on rockglacier creep (Kääb et al., 2007), the sensitivity of creep to temperature warming is smaller (a 1$^{o}$C warming results in only a 1.25 times increase in rate factor) and does not additionally increase towards warmer rockglacier temperatures (Fig. 4). Thus, the impact of warming on creep acceleration could be bigger than previously expected.

### 6.2.3 Adjustment timescales

The modeling shows for the faster and steeper Huhh1 rockglacier with about 100 a *adjustment times* of thickness and advance rates (Fig. 6d and 7d) that are an order of magnitude faster than the 1000 a of Murtèl rockglacier. While in the literature such differences in adjustment timescales have qualitatively been linked to the general rheology and mass turnover, the controlling factors remain rather unquantified. A comparison to theoretical considerations based on the kinematic wave theory developed by Nye (1963) can be made regarding the sediment supply experiment. The travelling wave speed $v_0$ of thickness changes is thereby given by

$$v_0 = (n + 2) \cdot u_s \tag{11}$$

which results in a typical adjustment timescale of this thickness change to reach the terminus of

$$T_k = \frac{L_0}{v_0} = \frac{L_0}{(n+2) \cdot u_s} \tag{12}$$

where $L_0$ is the length-scale of the rockglacier lobe. Thus, the timescale is inversely proportional to the horizontal velocity and consistent with our modeling means that a factor 10 difference in creep velocity between Huhh1 and Murtèl rockglacier translates into a factor 10 difference in adjustment time. For n=3, lengths of 300 m and 250 m and creep velocities of



0.07 ma$^{-1}$ and 0.6 ma$^{-1}$ for Murtèl and Huhh1, we obtain adjustment timescales of 860 a and 83 a respectively. These theoretical values agree in absolute and relative magnitude well with our modeled estimates. This implies that the kinematic wave speed, obtained from observed horizontal velocity, is a simple and meaningful measure for rockglacier adjustment times.

Based on the same theory (Nye, 1963), the diffusion of the thickness perturbation is proportional to the diffusivity which is given by

$$D_0 = \frac{n \cdot q}{\alpha} = \frac{n \cdot \bar{u} h}{\alpha} \qquad (13)$$

where $q$ is the ice flux and $\alpha$ the surface slope. For Murtèl and Huhh1 this gives diffusivities of 30 m$^2$a$^{-1}$ and 53 m$^2$a$^{-1}$, which are as a result of very small creep velocities and lower thicknesses much lower than obtained from a similar analysis on pure-ice glaciers (Eq. 13). Thickness perturbations spreading over length scales of $L_0$ result in diffusion timescales of

$$T_d = \frac{{L_0}^2}{D_0}. \qquad (14)$$

This results in 3000 a and 1200 a for the perturbation to spread over the whole landform for Murtèl and Huhh1, respectively, which is substantially longer than the timescales derived above for kinematic wave propagation. These diffusion timescales are also much longer compared to pure-ice glaciers and are consistent with the existence of the characteristic morphological features of ridges and furrows on the surface of rockglaciers. The very similar adjustment timescales obtained for all

perturbation experiments support the notion that the kinematic wave propagation timescale $T_k$ can be used as a general measure of adjustment in creep dynamics to a step change in external forcing. Note that the introduced adjustment timescale is, as a concept, similar to the *volume response time* for pure-ice glaciers for adjusting to a new climate (Johannesson, 1989) and should not be confused with a reaction time (time taken for a rockglacier to show a detectable reaction on an external forcing).

**6.2.4   Geometry change and subsidence**

Regarding geometry, the modeled rockglaciers respond to both warming and reduction in material input by a thinning of the landform that is fastest and most pronounced in the deposition area and in the upper parts of the rockglacier (Fig. 10). The front and therefore the landform as a whole remains advancing, although at slightly differing speeds depending on the applied perturbations (see Fig. 7b and 10a); even if the material input is completely stopped. The perturbation experiments

can also be compared with the observed subsidence data presented in Sect. 3, in order to investigate the potentially controlling mechanisms of such geometry change. The observations in Fig. 2 and 3 show a pronounced subsidence in the lower deposition area and upper parts of the rockglaciers of roughly -0.05 ma$^{-1}$ for the Murtèl rockglacier and -0.16 ma$^{-1}$ for



Huhh1 rockglacier while the lower rockglacier lobe shows relatively small or unchanging thicknesses and an advancing front. Thus, the spatial patterns in modeled geometry changes agree well with the observations. The absolute maximum rates of modeled subsidence are for both rockglaciers between 2 to 10 times smaller than the observed rates depending on the perturbation. These modeled rates result from an immediate warming of 1$^{o}$C of the entire rockglacier body which is a rather

extreme scenario. Further, the possible maximum rates of thinning from a total switch off in material input rates are limited to their pre-perturbation absolute values (0.006 ma$^{-1}$ for Murtèl and 0.022 ma$^{-1}$ for Huhh1), which are also almost an order of magnitude smaller than the observed subsidence rates.

Thus, we conclude that thinning due to thermally induced acceleration and a reduction in material input are not sufficient to explain observed subsidence patterns at Murtèl and Huhh1 rockglacier and that melt of subsurface ice is additionally

required for the observed volume loss. The process of landform thinning, especially in the upper parts of the rockglacier, has been described as a sign of degradation by Ikeda and Matsuoka, (2002b), Roer et al. (2008b) and Springman et al. (2013). However, subsidence through melt of subsurface ice remains poorly constrained through observations and our process understanding and models linking them to external forcing are still limited (Scherler et al., 2014).

## 7       Conclusions

This study uses a numerical flow model based on the conservation of mass within the cascading transport system of coarse debris to simulate the long-term and current evolution of rockglacier surface geometry and velocity. For a given sediment/ice input and rockglacier rheology, the model is able to generate observed rockglacier geometries and creep velocities in realistic timescales for two distinctly different rockglaciers. It is also capable of reproducing the continuing advancing front through creep which is often observed for rockglaciers (Barsch, 1996).

Climatic changes, especially increasing temperatures are expected to influence rockglacier dynamics in a profound way. Our modeling approach allows not only for investigation of the impact of a direct warming of the rockglacier material by adjusting the rheology (rate factor) but also for including the influence of changes in material input consisting of sediment and ice. Changes in geometry and related kinematics in response to such external perturbations can thereby be modeled and contribute towards a better understanding of the evolution of rockglaciers. Our detailed analysis of such perturbation

experiments and modeling sensitivity studies give the following insights on rockglacier dynamics:

-       Short-term changes in velocities and advance rates result from temperature variations whereas long-term geometric adaptations (thickness and advance rates) are mainly influenced by material supply. We show that a 1°C temperature increase in rockglacier temperature can result in a 1.5-3 time (150%-300%) acceleration of horizontal velocity depending on the initial thermal state of the rockglacier.

-       Both rockglacier temperature increase and reduction in material supply lead to thinning, while for the latter the maximum thinning rates are limited by the pre-perturbation material supply rate.





- Irrespective of the perturbation, the rockglaciers keep advancing and remain active although the thermal and sediment input conditions are not favorable for their sustenance which is consistent with field observations.

- Rockglaciers react spatially diverse to changes in environmental factors. Changes in temperature affect the entire landform immediately but the impact of material input variations are most pronounced in the sedimentation area and upper parts of the rockglacier. Comparing the model scenarios for localized topographic adaptations (subsidence) introduced by warming and variations in sediment/ice supply to observed subsidence features shows that these controlling factors are not sufficient to explain the magnitudes observed for our two examples. This implies that other processes such as melting of subsurface ice are responsible for subsidence and need further investigation.

- Although the absolute magnitudes in thinning and creep acceleration differ between the two rockglaciers, the changes relative to the initial thickness and creep velocity respectively are very similar thus indicating that changes scale with their geometric and dynamic characteristics.

- Based on most recent models of rockglacier rheology (Arenson and Springman, 2005), rockglaciers close to $0^{o}$C likely show much stronger reactions to thermal forcing than colder ones.

- On the basis of our modeling and kinematic wave theory, we propose a typical *timescale of dynamic adjustment* to external perturbations that is given by the inverse of a few times the horizontal velocity of a rockglacier. This timescale explains the order of magnitude difference in dynamic adjustment of our two chosen rockglacier examples which amount to 1000 a for Murtèl and 100 a for Huhh1.

The modeling approach presented here might serve as a useful tool to determine the dynamic state of alpine rockglaciers, their potential state of degradation and related forcing mechanisms. The growing amount of observations on geometric changes and rockglacier movements may thereby serve as important constraints for such model assessments and serve as indicators for the recent changes affecting periglacial high mountain systems. Therefore, future monitoring strategies should specifically be designed to detect spatially heterogeneous geometry changes and aim at observing entire slope systems, instead of focusing on single landforms.

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

**Tables to "Rockglacier on the move – Understanding rockglacier landform evolution and recent change from numerical flow modeling**

Table 1: Characteristics of the two selected rockglaciers (excluding their talus slopes)

|  | Murtèl rockglacier | Huhh1 rockglacier |
|---|---|---|
| Average thickness | 30 m | 12 m |
| Length | 280 m | 310 m |
| Slope | 12° | 27° |
| Age | ~ 5000- 6000 a | ~600 a |
| Hor. Velocity | 0.06 -0.13ma$^{-1}$ | 0.75 - 1.55 m a$^{-1}$ |
| Headwall area | 74687.1 m$^2$ | 82781.6 m$^2$ |
| Depositing area | 45931 m$^2$ | 32356 m$^2$ |

10    Table 2: Airborne and terrestrial remote sensing data available at the rockglacier sites.

| Data type | Murtèl | Huhh1 |
|---|---|---|
| **Airborne remote sensing (Photogrammetry and Airborne Laserscanning (ALS))** | RC30 in 1996 (PERMOS) RC30 in 2002 (PERMOS) ALS in 2003 RC30 in 2007 (PERMOS) | HRSC-A in 2001 (Otto el al. 2007) RC 30 in 2005 (Roer 2005) ALS in 2007 (Müller et al. 2014a) ADS 40 in 2010 (Müller et al. 2014a) ADS 80 in 2012 (Müller et al. 2014a) |
| **Geodetic Point Surveys** | 2009-2015 (annually) | 2001-2015 (annually) |

Table 3: Specific model input parameters for the rockglaciers. The sediment/ice input describes the volume of debris deposited on the accumulation area per year. The rate factor A is derived from Eq. 11 and the runtime of each rockglacier model is selected due to their approximated age (see Sect. 3).

| Input Parameter | Murtèl | Huhh1 |
|---|---|---|
| Material input rate | 0.006 m$^3$m$^{-2}$ a$^1$ | 0.022 m$^3$m$^{-2}$ a$^{-1}$ |
| Rate Factor A | 4.5*10$^{-18}$Pa$^{-3}$ a$^{-1}$ | 7*10$^{-17}$Pa$^{-3}$ a$^{-1}$ |
| Runtime | 6000a | 600a |
| Rockglacier Slope | 12$^o$ | 27$^o$ |





Table 4: Comparison of the observed (obs.) and modeled (mod.) rockglacier thickness and velocity for Murtèl and Huhh1 after 6000 years and 600 years respectively.

|  | **Murtèl obs.** | **Murtèl mod.** | **Huhh1 obs.** | **Huhh1 mod.** |
|---|---|---|---|---|
| **Length** | 280m | 300m | 310m | 240m |
| **Thickness** | 30m | 28m | 12m | 16m |
| **Horizontal velocity** | 0.06-0.13 ma$^{-1}$ | 0.06-0.09 ma$^{-1}$ | 0.75 - 1.55 ma$^{-1}$ | 0.63-0.79 ma$^{-1}$ |

Table 5: Multiplicative increase in rate factor A from a 1°C rockglacier warming for the different temperature relations of
5   Arenson and Springman (2005) and Paterson and Budd (1982) and varying rockglacier reference temperatures

| **Rockglacier reference temperature** | **Change in rate factor A for Arenson and Springman (2005) for a 1°C warming** | **Change in rate factor A for Paterson (1982) for a 1°C warming** |
|---|---|---|
| -2°C | 1.396*A | 1.254*A |
| -1.5°C | 1.705*A | 1.253*A |
| -1°C | 2.718*A | 1.252*A |



Table 6: All combined Experiments. The creep rate change is implemetented by increasing the rate factor A (Eq. 6) and the change in material accumulation by varying the accumulation rate $a_r$ (Eq. 9).

| Model run | Creep rate change | Accumulation change |
|---|---|---|
| 1.4*A and  0*Acc | 1.4*A | $0*a_r$ |
| 1.4*A and 0.4*Acc | 1.4*A | $0.4*a_r$ |
| 1.4*A and  1*Acc | 1.4*A | $1*a_r$ |
| 1.4*A and  2*Acc | 1.4*A | $2*a_r$ |
| 1.7*A and  0*Acc | 1.7*A | $0*a_r$ |
| 1.7*A and  0.4*Acc | 1.7*A | $0.4*a_r$ |
| 1.7*A and  1*Acc | 1.7*A | $1*a_r$ |
| 1.7*A and  2*Acc | 1.7*A | $2*a_r$ |
| 2.7*A and  0*Acc | 2.7*A | $0*a_r$ |
| 2.7*A and 0.4*Acc | 2.7*A | $0.4*a_r$ |
| 2.7*A and  1*Acc | 2.7*A | $1*a_r$ |
| 2.7*A and  2*Acc | 2.7*A | $2*a_r$ |



**Figures to "Rockglacier on the move – Understanding rockglacier landform evolution and recent change from numerical flow modeling**

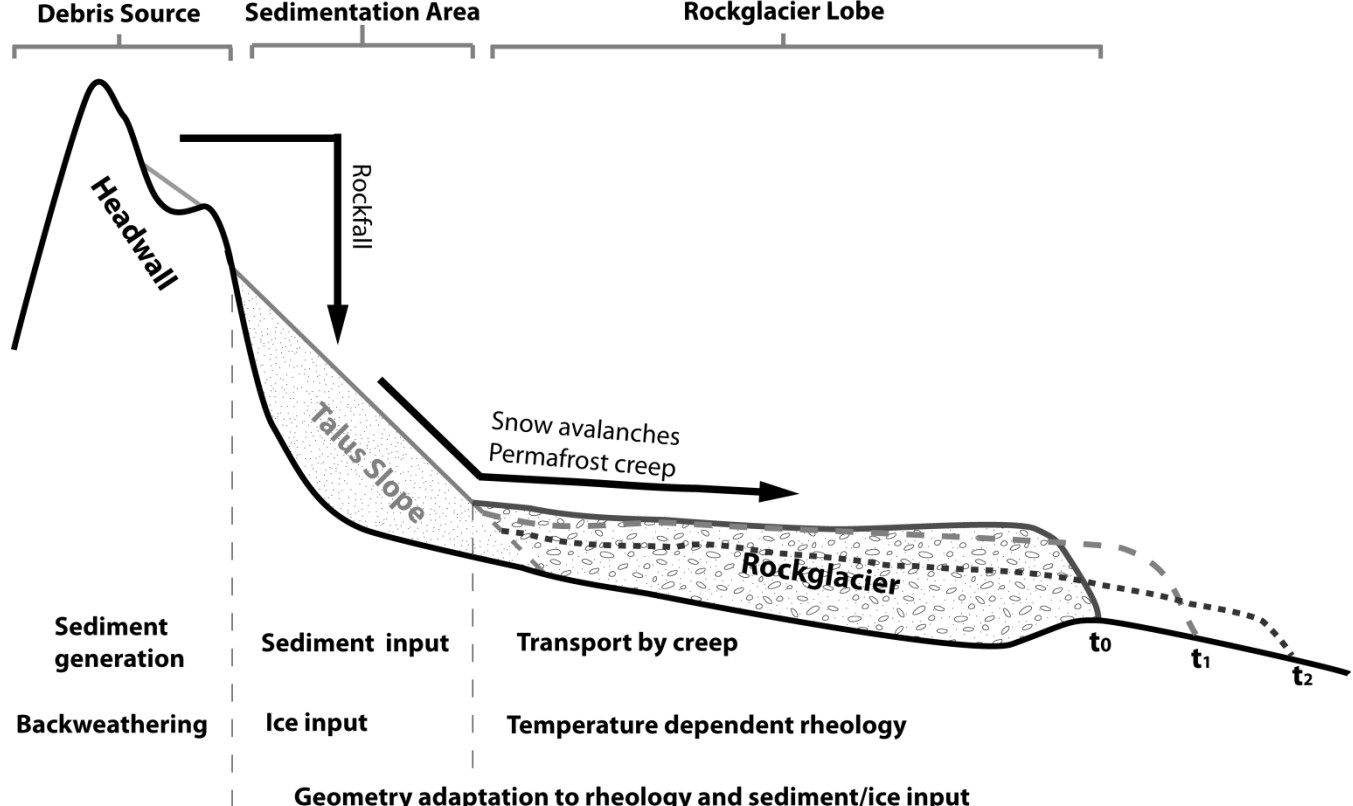

Figure 1: Conceptual model of the dynamic evolution of a rockglacier system (adapted from Fig. 1 in Müller et al., 2014a). Black arrows show the sediment transport. t0, t1 and t2 show the rockglacier surface geometries at different time-steps resulting from variations in environmental factors such as warming and a decrease of sediment/ice input.





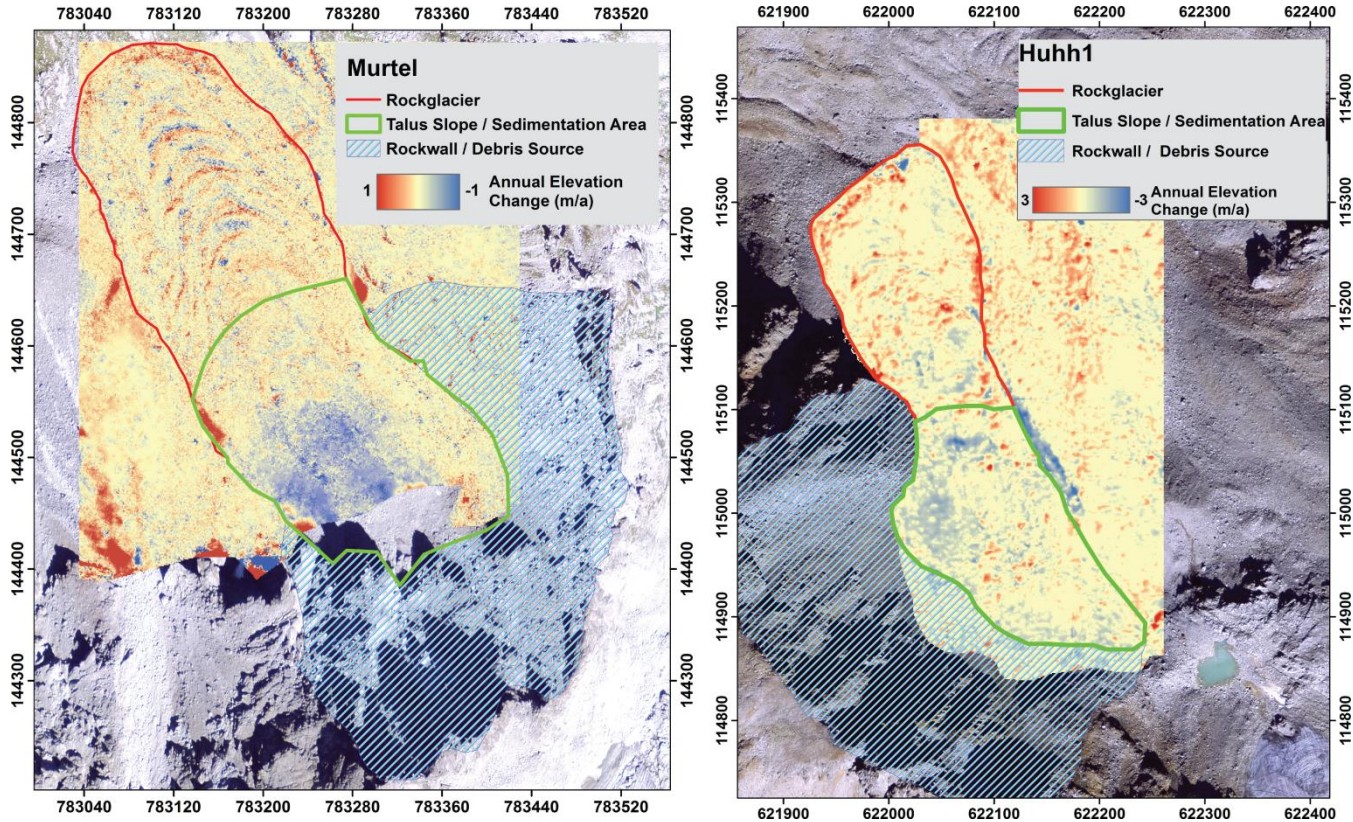

**Figure 2: The annual elevation change of the two rockglacier systems for Murtèl (period 1996-2007) and Huhh1 (period 2001-2012). The annual rates are derived from multisensoral and multitemporal remote sensing products (Tab. 2).**



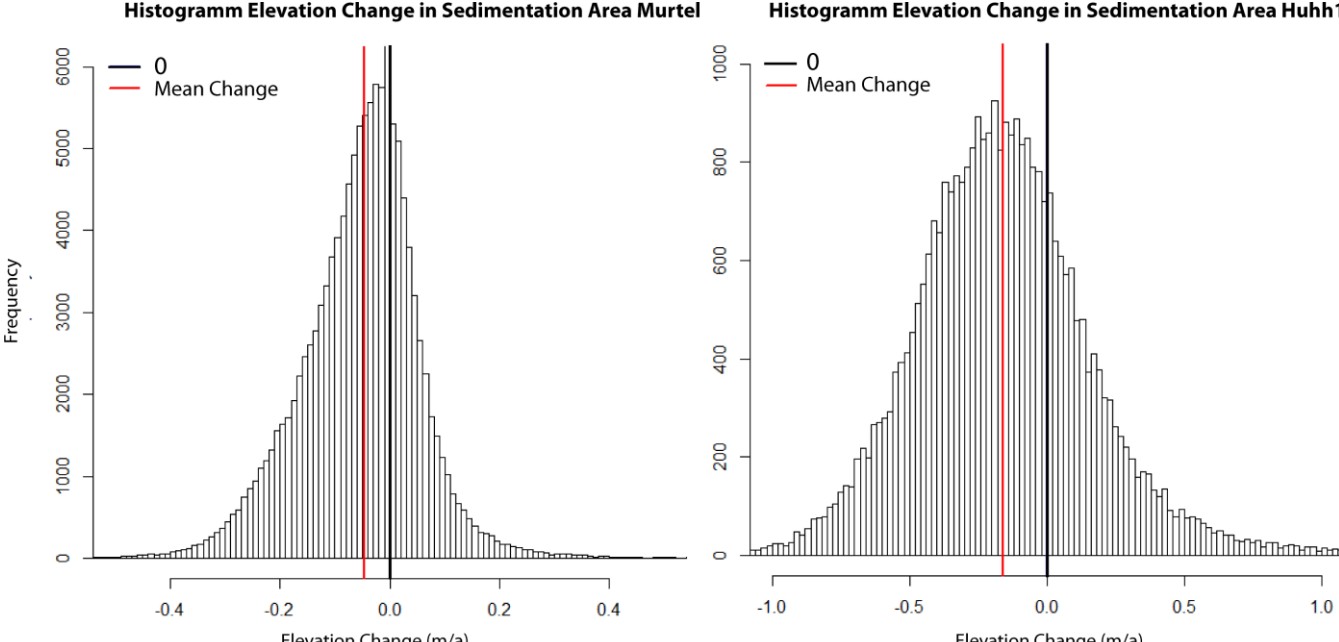

**Figure 3: The frequency distribution of annual vertical surface change (ma$^{-1}$) from DEM-differencing in the talus slope/ sedimentation area of the two rockglacier systems. Both systems show negative mean values (red lines). The black line refers to 0 ma$^{-1}$ difference.**





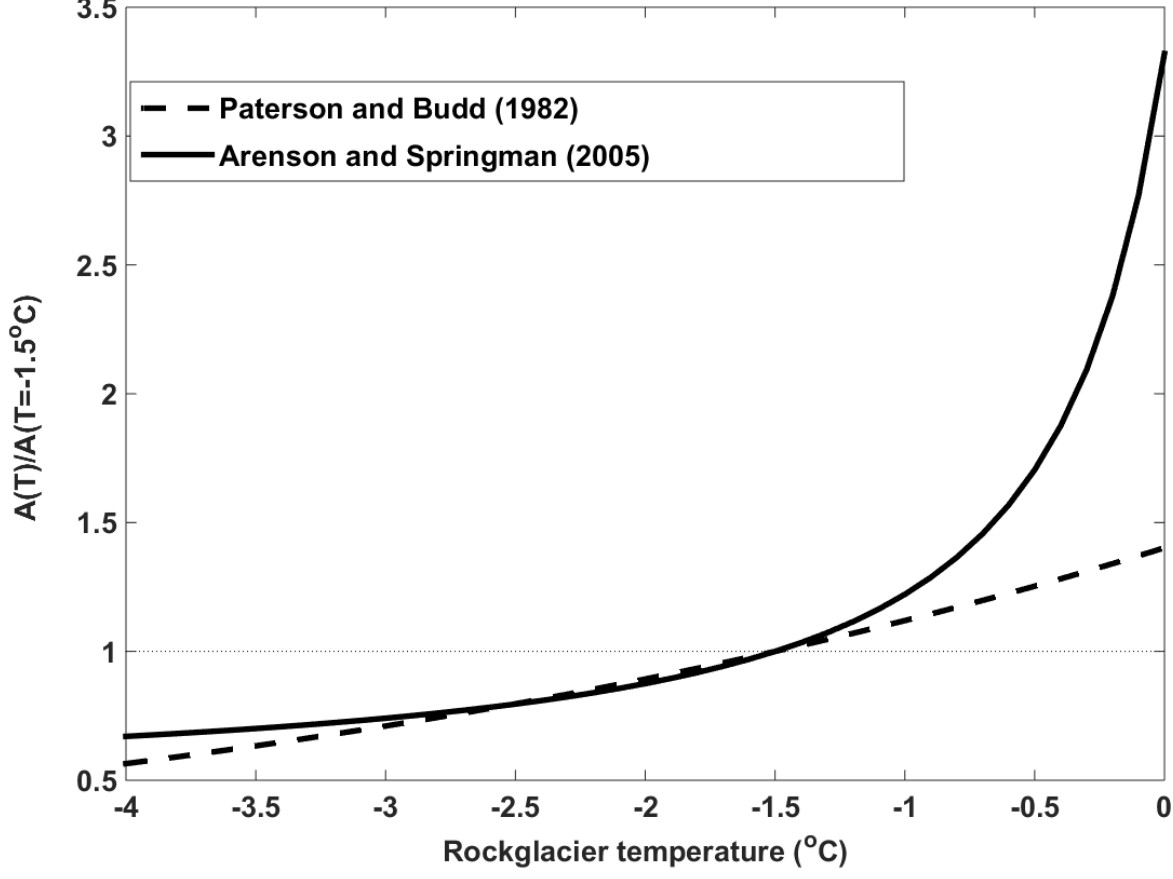

**Figure 4: Temperature dependence of the rate factor relative to the rate factor at a reference temperature of -1.5°C as derived for pure ice (Paterson and Budd, 1982) and for rockglacier material (Arenson and Springman 2005).**



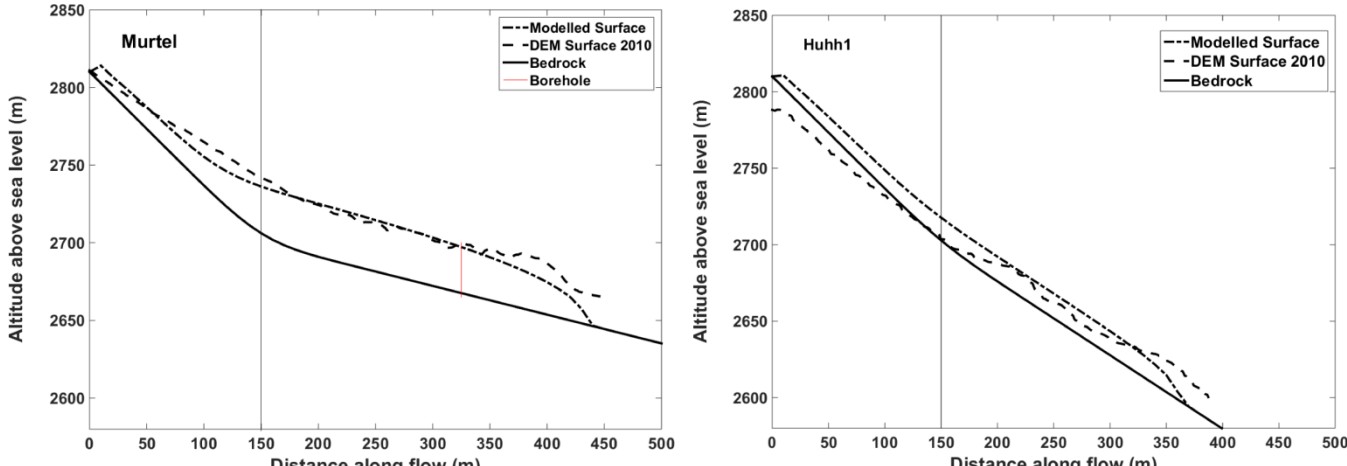

**Figure 5: Observed and modeled rockglacier geometry after build-up. The along flow bed topography used in the model and the modeled and observed (from DEMs) rockglacier surfaces are shown. The vertical fine black lines mark the boundary between the deposition area (talus slope) and the main rockglacier lobe.**





**Figure 6: Modeled evolution of surface geometry (a), absolute thickness (b), horizontal velocity (c) along the central flow line, terminus advance (d) and volume evolution (e) for the rockglacier build-up (first 6000a runtime) and for the successive temperature perturbation experiment (temperature increase of 1°C, with -1.5°C reference temperature). The black line shows the state of the system before the temperature step-change at 6000a. The lines are plotted at 100a time intervals.**





**Figure 7: Modeled evolution of surface geometry (a), absolute thickness (b), horizontal velocity (c) along the central flow line, terminus advance (d) and volume evolution (e) for Murtèl rockglacier when the material input is switched off at 6000a, after the rockglacier build-up (-1.5 C rockglacier temperature). The black line shows the state of the system before the switch-off of material supply at 6000years. The lines are plotted at 100yr time intervals.**



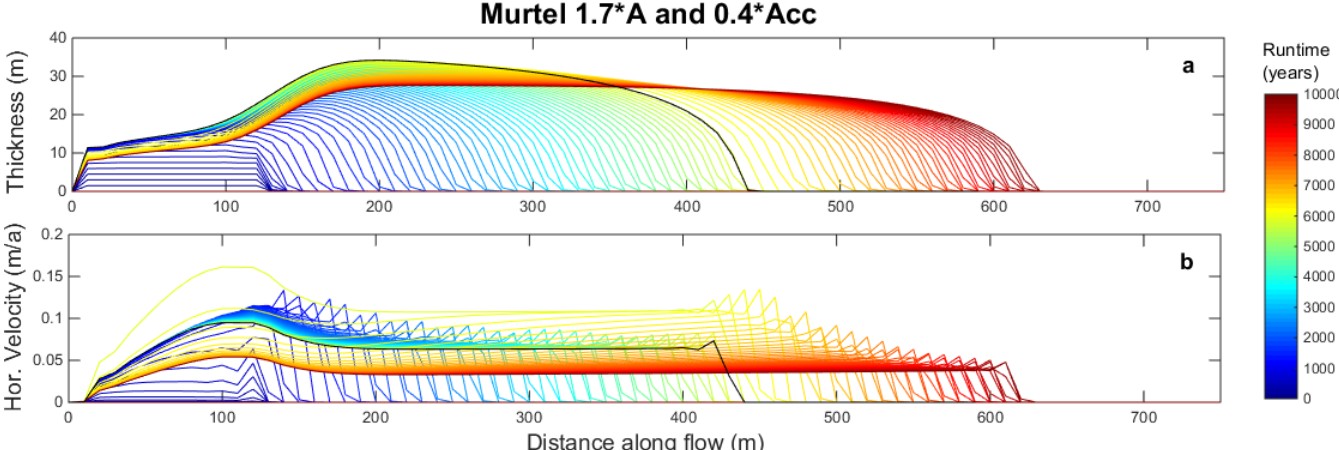

**Figure 8: Modeled evolution of absolute thickness (a) and horizontal velocities (b) of the Murtèl rockglacier introducing a 1°C temperature increase (1.7 times increase in rate factor) and a reduction in material input to 40% after the rockglacier build-up (6000a, black line). The black lines in all plots depict the state of the rockglacier as shown in figure 5 before the perturbations were introduced The lines are plotted at 100a time steps.**

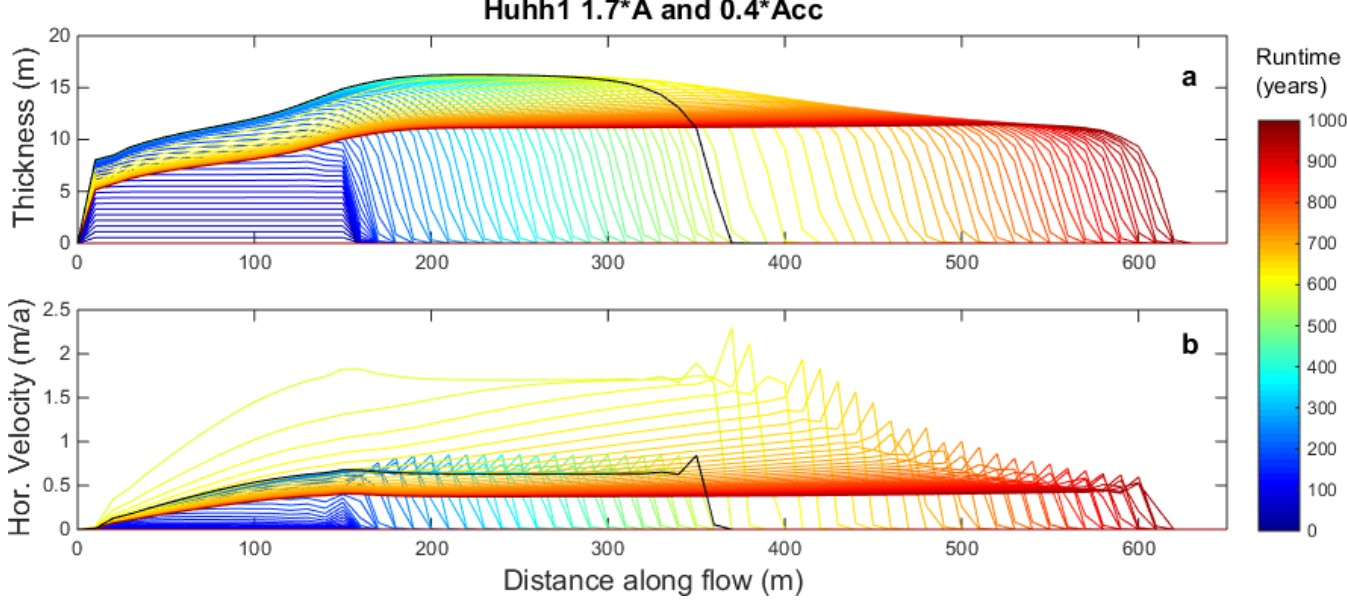

**Figure 9: The evolution of absolute thickness (a) and horizontal velocities (b) of the Huhh1 rockglacier introducing a 1°C temperature increase (1.7 times increase in rate factor) and 40% of the initial material input after rockglacier build-up (600a, black line). The black lines in all plots depict the state of the rockglacier as shown in figure 6 before the perturbations were introduced. The lines are plotted in 10a steps.**



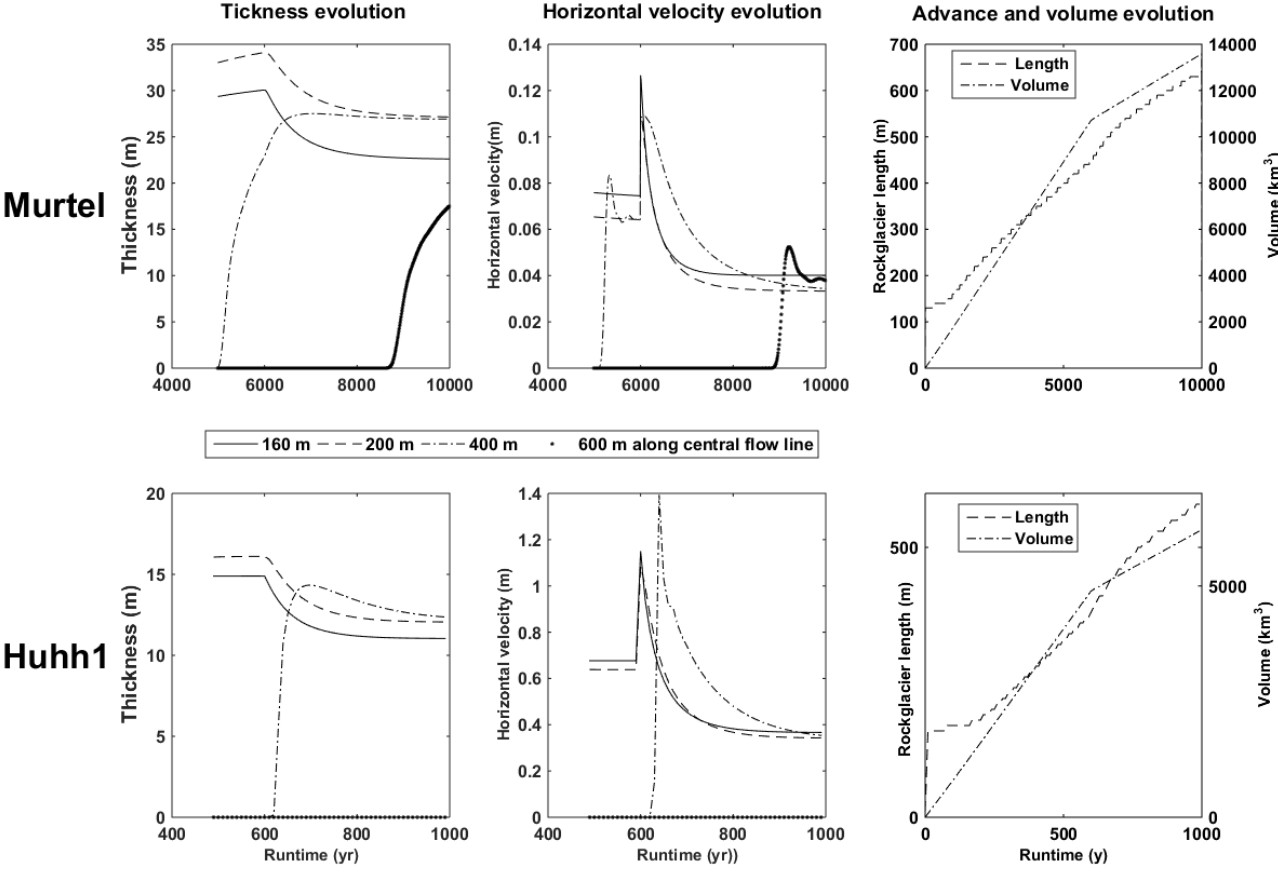

**Figure 10: Rockglacier evolution of thickness, horizontal velocity, advance and velocity after a 1°C temperature increase and 40% decrease in material input after rockglacier build-up. The dynamic evolution is shown for three points along the central flow line at 160m, 200m and 400m. The rockglacier is assumed to have an initial temperature of -1.5°C**





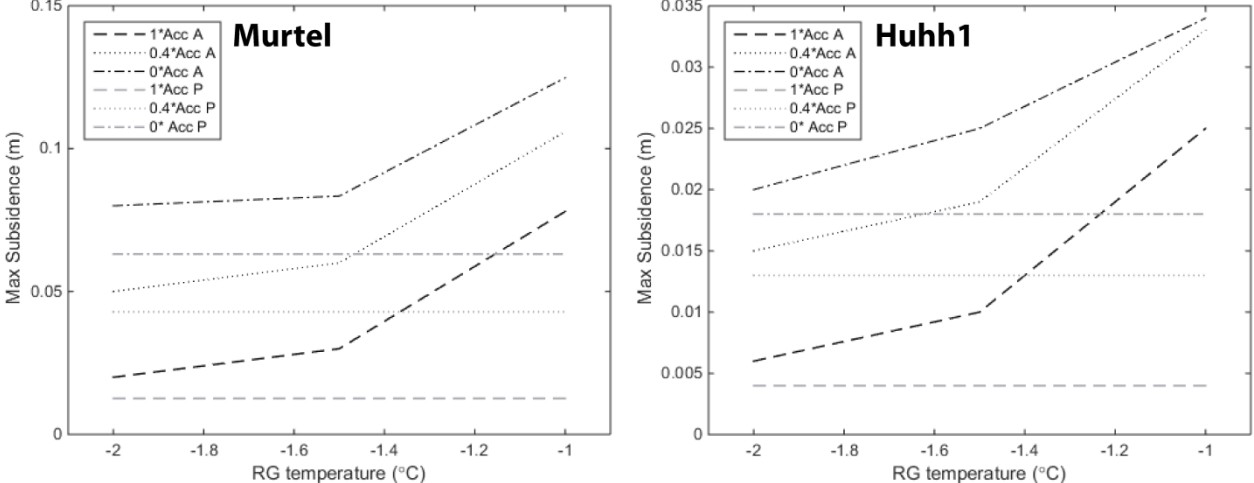

**Figure 11: Modeled maximum annual subsidence rates in the deposition area of the rockglaciers in relation to the reference temperature and change in material input and for the two temperature models of Arenson and Springman (2005; black lines) and Paterson and Budd (1982; grey lines) after a 1°C temperature increase. Note the different scales for subsidence for the two rockglaciers.**