# Peer review of "Rockglaciers on the run – Understanding rockglacier landform evolution and recent changes from numerical flow modeling"

_The Cryosphere, 2016_

## Referee Comment (RC1) · Lukas Arenson (Referee) · 1 May 2016

Dear authors,

I'd like to congratulate you on a very interesting paper that is well written and enjoyable to read. Even though you had to use various simplifications and make numerous assumptions, the overall findings are of great importance in improving our understanding of rock glacier dynamics. The paper is a first step into the right direction where dynamic modelling of rock glaciers could go. It is also refreshing to see that the authors understand the limitations of their approach and they do not try to over-interpret their results but take them for what they are: A first step.

[Figure]

I have attached an annotated version that has a couple of smaller comments that should be easy to address. One comments is related to the use of "rockglacier" instead of "rock glacier". I do understand the author's motivation, but I feel that this battle is lost and we should use the commonly used two-word term.

Kind regards, Lukas Arenson

Please also note the supplement to this comment:
http://www.the-cryosphere-discuss.net/tc-2016-35/tc-2016-35-RC1-supplement.pdf

**Supplement:**

[revised manuscript text omitted]

---

## Referee Comment (RC2) · C. Kinnard (Referee) · 20 May 2016

General comments

I really enjoyed reading this manuscript. The authors begin by analyzing and presenting field evidence for recent topographic changes observed on two rock glaciers in the Alps, which show thinning in their accumulation area over the last decade, and then go on to build a simple conceptual model of rock glacier dynamics to test the sensitivity of rock glacier dynamics to changes in temperature and debris-ice accumulation rate. The model sensitivity results are compared with observations and the authors conclude that changes in rock glacier temperature and material (debris and ice) influx are not sufficient to explain the observed thinning rates - hence internal ice melting must also

contribute to thinning. This is a classical and very elegant approach to scientific testing and I congratulate the authors for this nice work. I also concur with the other reviewer - that you have been careful in not over-interpreting your model, which does includes simplifications. However some of my specific comments will suggest discussing more thoroughly the possible implication of these simplifications on the results.

I have mainly minor comments which are addressed below and small technical corrections annotated in the uploaded manuscript.

Specific comments

Terminology

- rock glacier VS rockglacier: I believe the former is correct.

DEM differencing accuracy

P.5 L20: "The limitations concerning processing, uncertainties and application are presented in Kääb and Vollmer (2000), Roer et al. (2005c), Roer and Nyenhuis (2007) and Müller et al. (2014b), and applied in this assessment."

-My first question when looking at the DEM difference field in Figure 2 is what is the error on dZ/dt? In other words what is the detection limit? If this information has indeed been produced before you should at least state it, in the text or in the figure caption, for the benefit of the reader.

- My other question is why not presenting the average dZ/dT for the whole rock glacier surface as well? This should give you the total (or average) rock glacier mass-balance, assuming that the data is precise enough: if it is negative then internal ice melting has been larger than mass influx (sediments + ice). You present contrasting results (p6, L10) on the 'main lobes' where DEM differences and survey data do not give consistent results. The difference may be due to sampling errors in the survey data (22 points if I recall), or else DEM errors which are not presented.

[Figure]

Rheology of ice-debris mixture and the shallow ice approximation

- The viscosity (or the softness parameter A) depends on temperature, but also on fabric, and moisture content. The later is particularly true for 'warm' rock glaciers , i.e. close to their melting point. For 'warm' ice increasing debris concentration may in fact decrease the viscosity by favouring higher moisture content at ice-debris interfaces, up to a critical debris concentration past which inter-particle frictional strength increases the viscosity (see reference below and other works cited in them). I do not expect that you include the sensitivity to moisture in your model, but that this aspect be included in the literature review on rock glacier rheology, and discussed with respect to its implication for the sensitivity of the softness parameter to temperature as climate warms. An excellent review of the rheology of debris-ice mixture is also given by Moore (2014), which you should refer to in your introduction and discussion.

Moore PL. 2014. Deformation of debris−ice mixtures. Review of Geophysics 52: 435-467.

Monnier, S., & Kinnard, C. (2016). Interrogating the time and processes of development of the Las Liebres rock glacier, central Chilean Andes, using a numerical flow model. Earth Surface Processes and Landforms.

Rock glacier mass-balance

(p.8 L25) : The rock glacier density is kept constant in space and time in your model, you should state it clearly in the model definition. This means that (i) no internal ice melting can take place, and (ii) that variations in active-layer thickness, which in rock glacier is often blocky and dry, are not considered. Hence in your model experiment the rock glacier is allowed to grow downslope indefinitely over time. In reality as the rock glacier front flows downslope/down-valley warmer temperature there should increase internal ice melting, i.e. the active layer should deepen, which could have a stabilizing effect on flow by increasing the frictional resistance. This is in my opinion the most important oversimplification of the model: you consider mass influx (debris+ice with a

fixed fraction) but mass outflux is not permitted. See Konrad and Humphrey (2000) for an example of prescribed mass balance profile on rock glaciers (debris-covered glacier in their case). The active layer will isolate the internal ice from the atmosphere but climate warming on decadal to centennial scales will change the active-layer thickness. This point should be emphasized and the implications discussed. Including mass loss could be a nice addition for follow-up work.

Konrad, S. K., & Humphrey, N. F. (2000). Steady-state flow model of debris-covered glaciers (rock glaciers). IAHS PUBLICATION, 255-266.

Sallow ice approximation and side friction in a parabolic cross-valley profile

P.10 L8: you use the shallow ice approximation which implies a slab of infinite width, i.e. without friction effects, yet you parameterise the rock glacier bed with a parabolic cross-valley profile of finite width. Consider using a side drag parameterisation (shape factor, see e.g. Cuffey and Paterson, 2010, and use in Monnier and Kinnard, 2016). If you do not include it, discuss why not and possible effects of ignoring side friction: the driving dress can be reduced by 10-25% depending on the cross-profile shape (this if the shape factor approach is valid).

Combined sediment and temperature sensitivity experiments (section 5.2.3.)

(p15 L7, and Figure 10): Why did you choose to present the model run with a decrease of material input to 40% of the original value? I may have missed it: is it because this particular combination fits the observation better in some way? It is not clear and could be better presented. Do you have a basis to assume that there has been a 60% decrease in mass influx to the rock glacier in the past?

Section 6.2.3 Adjustment times

I found this section somewhat confusing, especially with regards to the analogy with glacier response times definitions (p20 L21), and the section deserves some clarification.

i) Your adjustment time scale is based on the kinematic wave speed (cf equation 12) . This is the time taken for the topographic perturbation to travel to the front, and is faster than the ice velocity.

ii) Your diffusion timescale (cf equation 14) is the time taken for the topographic perturbation to diffuse over the whole landform, and is longer than the adjustment time.

My impression is that your adjustment time (kinematic wave) is more akin to the reaction time that you mentioned for glaciers, i.e. the time it takes for the front to respond to a mass-balance perturbation, while your diffusion time scale is in turn more akin to the volumetric time scale, i.e. the time taken for (most of) the landform to adjust to a new mass-balance state. You say the reverse. I may get this wrong, but you may want to clarify better theses concepts and the analogies to glaciers. The fact that the terminology of response times in the glaciology literature is somewhat confused certainly does not help... See discussion in Hooke (Principles of Glacier Mechanics, 2005, p.375) for example for the different definitions.

P.7 L6 : "Surface features (e.g. slope, substrate) as well as velocity fields are further used to delimit the different subsystems."

- Which velocity fields? If you refer to previously published data insert the proper reference.

Please also note the supplement to this comment:
http://www.the-cryosphere-discuss.net/tc-2016-35/tc-2016-35-RC2-supplement.pdf

[Figure]

**Supplement:**

[revised manuscript text omitted]

---

## Author Comment (AC1) · 8 Jul 2016

**Author response to reviewer comments**

In the following we respond to the reviewers comments on our document. Since they raise similar points we reply to both reviewers in one document. C. Kinnard (Reviewer 2) stated some specific comments in his reply which we respond to (R2.x) before we reply to the all comments in the manuscript of L. Arenson(reviewer 1) (M1.x) and C. Kinnard (M2.x) posted directly into the manuscript.

Note for notation in our response below: In blue our response to the reviewers comments and italics marks the text as it has been changed to in the revised paper.

**1    Specific comments Reviewer C Kinnard (Reviewer 2)**

| | |
|---|---|
| R2.1 | Terminology

- rock glacier VS rockglacier: I believe the former is correct.

We would like to hold on to the term rockglacier On the one hand, this study follows the periglacial reasoning, and in order to emphasize the autonomy of the phenomenon, the term 'rockglacier' is written in one word, following Barsch (1988). Thus, rockglaciers are significant indicators of past and present permafrost distribution and related climatological changes.

On the other (and the more pragmatic) hand, the first author has used the term 'rockglacier' in all his previous publications and it would introduce a break of consistency in his work and dissertation of which this publication is the final part of. |
| R2.2 | P.5 L20: "The limitations concerning processing, uncertainties and application are presented in Kääb and Vollmer (2000), Roer et al. (2005c), Roer and Nyenhuis (2007) and Müller et al. (2014b), and applied in this assessment." |
| R2.3 | -My first question when looking at the DEM difference field in Figure 2 is what is the error on dZ/dt? In other words what is the detection limit? If this information has indeed been produced before you should at least state it, in the text or in the figure caption, for the benefit of the reader.

We corrected the systematic error/trueness (Mentitto et al. 2007) of the remote sensing data using the terrestrial survey points (TS, tachymeter) as reference for the TLS and the TS and ALS data as reference for the photogrammetric DEM data.

Therefore we were able to set a detection limit of 20 cm for the ALS and a slope dependent detection limit derived from the assessment presented in Mueller et al. 2014b. The slope dependent error for the photogrammetric error ranges from 0.5m to 0.9m.

Since we use multiple independent datasets with different precisions, we are confident that we can use them to detect even such small topographical changes. Winsvold et al. (2016) have shown that by 'stacking and investigating larger time series for individual pixels, … (they are) able to reduce noise and the chance for bias, compared to using single observations.' We applied a similar approach where we checked the annual subsidence rates between all individual |

timesteps of all sensor systems (ALS vs Tachymeter vs Photogrammetry)to the longest timespan (which has the clearest signal) and ended up with very similar annual subsidence rates which we used for our analysis.

To clearify we changed in the manuscript:

*The systematic error/trueness (Menditto et al. 2007) of the remote sensing data was corrected using the terrestrial geodetic survey points (tachymeter) as reference for the ALS and the ALS data as reference for the photogrammetric DEM data as presented in Müller et al. (2014b). The spatial coverage of each reference dataset allowed to establish a slope dependent detection limit derived from the assessment ranging from 0.5 m to 0.9 m (see also Mueller et al. 2014b). The availability of numerous multisensoral datasets enabled stacking, investigating dense time steps and crosschecking products with different precisions (terrestrial survey, ALS, photogrammetry) which led to a high reliability of the derived annual subsidence rates.*

And added the following references:

*Menditto, A., Patriarca, M., and Magnusson, B.: Understanding the meaning of accuracy, trueness and precision, Accred Qual Assur, 12, 45–47, doi:10.1007/s00769-006-0191-z, 2007.*

*Winsvold, S. H., Kaab, A., and Nuth, C.: Regional Glacier Mapping Using Optical Satellite Data Time Series, IEEE J. Sel. Top. Appl. Earth Observations Remote Sensing, 1–14, doi:10.1109/JSTARS.2016.2527063, 2016.*

| | |
|---|---|
| R2.4 | - My other question is why not presenting the average dZ/dT for the whole rock glacier surface as well? This should give you the total (or average) rock glacier mass-balance, assuming that the data is precise enough: if it is negative then internal ice melting has been larger than mass influx (sediments + ice). You present contrasting results (p6, L10) on the 'main lobes' where DEM differences and survey data do not give consistent results. The difference may be due to sampling errors in the survey data (22 points if I recall), or else DEM errors which are not presented.

After further consideration, we decided to omit the subsidence analysis derived from the terrestrial survey. Since these surveying points were originally installed for the monitoring of rockglacier movement, they are mounted on selected boulders which are exceptionally stable and deeply embedded in the rock-ice matrix and might therefore not be useful for the analysis of surface subsidence. To reduce confusion we therefore deleted the corresponding paragraph.

A closer look at the annual subsidence over the entire rockglacier area shows a general thinning of the entire landform (0.02 ma-1 for the Murtèl rockglacier and -0.09 ma-1 for the Huhh1 rockglacier (see Fig.1 in this document for the corresponding histograms) derived from the multisensoral analysis. The magnitude of subsidence in the deposition area is is almost almost twice as high as the overall average annual subsidence of the landforms and already just the visual inspection of the DEM differencing (Fig. 2 in the manuscript) shows the thinning signal. Therefore we think that the focus of the subsidence analysis should remain on the deposition area (and therefore Fig 3) due to its strong signal and previous mentioning in the literature. |

[Figure]

Fig. 1: The frequency distribution of annual vertical surface change (ma$^{-1}$) from DEM-differencing over the entire rockglacier systems. Both systems show negative mean values (red lines). The black line refers to 0 ma$^{-1}$ difference.

We included the subsidence values for the entire landforms in the manuscript with the corresponding paragraph reading as:

*This analysis (over decadal time periods) showed distinct subsidence features of different magnitudes on the entire rockglacier, most pronounced in the deposition area (outline with green in Fig. 2) of the rockglacier. A more detailed assessment of the subsidence shows a general lowering of the entire surface of the rockglacier system of 0.02 ma$^{-1}$ for the Murtèl rockglacier and -0.09 ma$^{-1}$ for the Huhh1 rockglacier. The spatial analysis of the subsidence phenomenon depicts the strongest signal of annual lowering in the deposition area (talus slope/sedimentation area, see Fig. 3) with an overall average of annual subsidence of -0.04 ma$^{-1}$ for the Murtèl rockglacier and -0.16 ma$^{-1}$ for the Huhh1 rockglacier. Such subsidence features, especially in the deposition area, have been described as signs of permafrost degradation (Roer et al., 2008a, Springman et al., 2013, Bodin et al., 2015) and are assessed by the rockglacier evolution model in Sect. 6.2.4.*

| | |
|---|---|
| R2.5 | - The viscosity (or the softness parameter A) depends on temperature, but also on fabric, and moisture content. The later is particularly true for 'warm' rock glaciers , i.e. close to their melting point. For 'warm' ice increasing debris concentration may in fact decrease the viscosity by favouring higher moisture content at ice-debris interfaces, up to a critical debris concentration past which inter-particle frictional strength increases the viscosity (see reference below and other works cited in them). I do not expect that you include the sensitivity to moisture in your model, but that this aspect be included in the literature review on rock glacier rheology, and discussed with respect to its implication for the sensitivity of the softness parameter to temperature as climate warms. An excellent review of the rheology of debris-ice mixture is also given by Moore |

(2014), which you should refer to in your introduction and discussion.

*We agree with the reviewer that moisture content is influencing the rheology and the rate factor specifically. We do not explicitly include changing in moisture content in our rheology model but this is implicitly included over the temperature. We added some explanation to this and the two suggested references, with the text now reading as:*

*The rate factor A is estimated from observed surface flow speeds by inverting Eq. (3) for A but is known to be influenced by the material temperature but also by other factors such as the moisture and debris content (Cuffey and Peterson, 2010). Moisture content is known to vary with time and temperature (see Moore, 2014 and Monnier and Kinnard, 2016) but we do not explicitly include this in our model as such changes are poorly constrained. The temperature related effect from moisture is however implicitly included by writing the rate factor of the rockglacier material as a product of the temperature dependent part $A^*(T)$ and a scaling factor $f_A$ accounting for the influence of the debris:*

$$A = A^*(T) \cdot f_A \qquad\qquad (6)$$

*This approach is in agreement with known rheological investigations (Paterson and Budd, 1982, Arenson and Springman, 2005) and allows including a temperature forcing in our modelling experiments.*

| R2.6 | (p.8 L25): The rock glacier density is kept constant in space and time in your model, you should state it clearly in the model definition. This means that (i) no internal ice melting can take place, and (ii) that variations in active-layer thickness, which in rock glacier is often blocky and dry, are not considered. Hence in your model experiment the rock glacier is allowed to grow downslope indefinitely over time. In reality as the rock glacier front flows downslope/down-valley warmer temperature there should increase internal ice melting, i.e. the active layer should deepen, which could have a stabilizing effect on flow by increasing the frictional resistance. This is in my opinion the most important oversimplification of the model: you consider mass influx (debris+ice with a fixed fraction) but mass outflux is not permitted. See Konrad and Humphrey (2000) for example of prescribed mass balance profile on rock glaciers (debris-covered glacier in their case). The active layer will isolate the internal ice from the atmosphere but climate warming on decadal to centennial scales will change the active-layer thickness. This point should be emphasized and the implications discussed. Including mass loss could be a nice addition for follow-up work.

Konrad, S. K., & Humphrey, N. F. (2000). Steady-state low model of debris-covered glaciers (rock glaciers). IAHS PUBLICATION, 255-266.

*We used a constant rock glacier material density and clarified this in the text.*

*We further agree with the reviewer that internal ice melt may occur but its relationship to external temperature forcing is very poorly constrained by data and models and elevation differences between the tongue and the upper part of the rock glacier in our case very small. We therefore purposely excluded this process in our modelling experiments in this paper (out of* |
| --- | --- |

scope). We are actually already are working on some further research in this direction.

To clarify these point we added this in the text in section 4.2 Rockglacier creep modelling approach:

*For simplification we assume the rockglacier material to be a homogenous mixture of ice and sediment, meaning the rheological parameters such as the rate factor $A$, flow exponent $n$ and rock glacier material density $\rho_r$ do not change within the rockglacier body. This means that we exclude any internal ice melting and consider the active layer rheologically in the same way as the rock glacier material.*

…and at the of the discussion section 6.2.4 (geometry change and subsidence)

*…and therefore we did not include this process in our flow modelling. In future work, ice melt (as a function of temperature) could technically be easily included through a negative accumulation term in the surface evolution equation (8) and an adjustment of the mean density.*

| R2.7 | P.10 L8: you use the shallow ice approximation which implies a slab of infinite width, i.e. without friction effects, yet you parameterise the rock glacier bed with a parabolic cross-valley profile of finite width. Consider using a side drag parameterisation (shape factor, see e.g. Cuffey and Paterson, 2010, and use in Monnier and Kinnard, 2016). If you do not include it, discuss why not and possible effects of ignoring side friction: the driving dress can be reduced by 10-25% depending on the cross-profile shape (this if the shape factor approach is valid).

Our across glacier parabolic shape is actually included to account for an adjustment in width with varying thickness (in particularly at front). It has no impact on the flow speed. Yes we could of course include a shape factor but this would not change our modelling results at all our channel shape is constant along flow and the shape factor would implicitly be included in the scaling factor f_A for the rate factor (eqn 6) and thus make no difference. We added a sentence to explain this:

*This allows the width of the rock glacier to vary with changing thickness, but the effect of side drag is not explicitly included, but implicitly is contained in the scaling factor $f_A$ of the rate factor.* |

| R2.8 | (p15 L7 and Figure 10): Why did you choose to present the model run with a decrease of material input to 40% of the original value? I may have missed it: is it because this particular combination fits the observation better in some way? It is not clear and could be better presented. Do you have a basis to assume that there has been a 60% decrease in mass influx to the rock glacier in the past?

We introduce a 60% reduction in material influx because we assume that an expected climate warming will most strongly influence the aggradation and incorporation of subsurface ice. We agree that the studies on rockwall dynamic and their future activity hint towards an increase in |

sediment production on the short but the long-term prediction is rather uncertain. Krautblatter et al. 2013 show that the warming and subsequent thawing permafrost will increase instabilities in the rockwalls but 'dry' and ''warm' rockwalls will become more stable again on the long term. Therefore we assume for the combined experiment that the material input from the rockwall will remain the same but the ice input will change significantly. A 60% reduction of ice due to a 1°C warming is of course extreme but it exemplifies the impact of such potential environmental changes.

This argumentation is indeed not very clear in the manuscript therefore we added the following text and reference:

*We assume that warming and subsequent thawing of permafrost in rockwalls will lead on the short-term to an increase in sediment production but is expected to extenuate on the long-term (Krautblatter et al., 2013). Therefore we keep the material input from the rockwall constant but reduce the overall material influx by the amount of ice.*

*Krautblatter, M., Funk, D., and Günzel, F. K.: Why permafrost rocks become unstable: a rock–ice-mechanical model in time and space, Earth Surf. Process. Landforms, 38, 876–887, doi:10.1002/esp.3374, 2013.*

| | |
|---|---|
| R2.9 | I found this section somewhat confusing, especially with regards to the analogy with glacier response times definitions (p20 L21), and the section deserves some clarification.

i) Your adjustment time scale is based on the kinematic wave speed (cf equation 12). This is the time taken for the topographic perturbation to travel to the front, and is faster than the ice velocity.

ii) Your diffusion timescale (cf equation 14) is the time taken for the topographic perturbation to diffuse over the whole landform, and is longer than the adjustment time. My impression is that your adjustment time (kinematic wave) is more akin to the reaction time that you mentioned for glaciers, i.e. the time it takes for the front to respond to a mass-balance perturbation, while your diffusion time scale is in turn more akin to the volumetric time scale, i.e. the time taken for (most of) the landform to adjust to a new mass-balance state. You say the reverse. I may get this |

wrong, but you may want to clarify better theses concepts and the analogies to glaciers. The fact that the terminology of response times in the glaciology literature is somewhat confused certainly does not help... See discussion in Hooke (Principles of Glacier Mechanics, 2005, p.375) for example for the different definitions.

We agree that this section was probably slightly confusing, in particular together with the reference to the 'response time' of glaciers. Because we have no surface ice melt in our model, our adjustment time is very different to the classic Johannesson Volume response time scale for glaciers (which requires the ablation at the tongue). As the proposed adjustment time scale is easy to estimate from flow speed we think this measure may be very useful for dynamic assessments of rock glaciers but requires a proper explanation.

We therefore rewrote and restructured this section completely and consistent with Hooke (2010), and made much clearer that the proposed 'adjustment time' scale is NOT related to the glacier response time. The whole paragraph no reads as:

*The modeling shows that the adjustment times in thickness and advance rates are for the faster and steeper Huhh1 rockglacier with 100 a, an order of magnitude faster than the 1000 a of Murtèl rockglacier (Fig. 6d and 7d). While in the literature such differences in adjustment timescales have qualitatively been linked to the general rheology and mass turnover, the controlling factors remain unquantified. A comparison to theoretical considerations based on the kinematic wave theory developed by Nye (1963) can be made for the material supply experiment. Besides the change in material supply rate, the thickness adjustment depends mainly on the travelling wave speed of the thickness perturbation and on its diffusion. According to Nye (1963; and see also Hooke 2005, p. 371-381) the travelling wave speed $v_0$ is given by a multiple of the the rock glacier flow speed and specifically*

$$v_0 = (n + 2) \cdot u_s \qquad\qquad (11)$$

*The diffusion of the thickness perturbation is proportional to the diffusivity which is given by*

$$D_0 = \frac{n \cdot q}{\alpha} = \frac{n \cdot \bar{u} h}{\alpha} \qquad\qquad (12)$$

*where q is the ice flux and $\alpha$ the surface slope. For Murtèl and Huhh1 this results in diffusivities of 30 $m^2 a^{-1}$ and 53 $m^2 a^{-1}$, which are as a result of very small creep velocities and low thicknesses much lower compared to diffusivities of pure-ice glaciers.*

*Following Johannesson et al. (1989; see also Hooke, 2010, p. 376) the related 'adjustment time-scales' of thickness changes to expand over the whole rock glacier lobe then depend on a propagation time-scale $T_p$ and a diffusion time-scale $T_d$. The propagation time-scale is given by*

$$T_p = \frac{L_0}{v_0} = \frac{L_0}{(n+2) \cdot u_s} \qquad\qquad (13)$$

*where $L_0$ is the length-scale of the rockglacier lobe. This timescale is inversely proportional to the horizontal velocity and consistent with the modeling results, that a factor 10 difference in creep velocity between Huhh1 and Murtèl rockglacier translates into a factor 10 difference in adjustment time. For n=3, lengths of 300 m and 250 m and creep velocities of 0.07 $ma^{-1}$ and 0.6 $ma^{-1}$ for Murtèl and Huhh1, we obtain adjustment timescales of 860 a and 83 a respectively. These theoretical values agree in absolute and relative magnitude well with our modeled estimates.*

| | |
|---|---|
| | *The diffusion timescale is given by*

$$T_d = \frac{{L_0}^2}{D_0}. \qquad\qquad\qquad (14)$$

*which results in 3000 a and 1200 a for Murtèl and Huhh1, respectively, for the thickness perturbation to spread over the entire landform and is substantially longer than the timescales derived above for the kinematic wave propagation. These diffusion timescales are also much longer compared to pure-ice glaciers and are consistent with the existence of the characteristic morphological features of ridges and furrows on the surface of rockglaciers.*

*To conclude, the propagation timescale, and therefore the, horizontal velocity, is a simple and meaningful measure for rockglacier 'adjustment times'. The very similar adjustment times obtained for the different types of perturbation experiments support the notion that this propagation timescale $T_p$ can be used as a general measure of adjustment in creep dynamics to a step change in external forcing. Note that the introduced adjustment timescale is not the same as the 'volume response time' for pure-ice glaciers for adjusting to a new climate (Johannesson, 1989) and should not be confused with a 'reaction time' (time taken for a rockglacier to show a detectable reaction on an external forcing).* |
| R2.10 | P.7 L6: "Surface features (e.g. slope, substrate) as well as velocity fields are further used to delimit the different subsystems." - Which velocity fields? If you refer to previously published data insert the proper reference.

We calculated velocity fields for both rockglaciers from feature tracking in TLS and photogrammetric data but refrained from showing it in another figure since it would introduce another figure and the benefit from said figure would be marginal.

Similar but older and less comprehensive data have been shown in Roer et al., 2005 and Kääb et al., 1998.

In order to clarify we now reference

*Roer, I., Kääb, A., and Dikau, R.: Rockglacier kinematics derived from small-scale aerial photography and digital airborne pushbroom imagery, Zeitschrift für Geomorphologie, 49, 73–87, 2005.*

*Kääb, A., Gudmundsson, G. H., and Hoelzle, M.: Surface deformation of creeping mountain permafrost. Photogrammetric investigations on rock glacier Murtel, Swiss Alps., in: 7th International Permafrost Conference Proceedings, Yellowknife, USA, 531–537, 1998.* |

**2   Response to Comments in Manuscript L. Arenson (Reviewer 1)**

| | |
|---|---|
| M1.1 | Page: 1 Text: I would prefer to use "rock glacier" in 2 words instead of 1 word. I do understand the motivation behind using a single term, but this is a fight against windmills in my view. Using |

| | the two word version makes the paper easier to be found during general search by most people.

See R2.1 |
|---|---|
| M1.2 | Page: 1  Highlight: Indicative of permafrost creep Text: I'd say they are geoforms that form as a result of permafrost (actually excess ground ice, since permafrost is only define by temperature and time) creep and not indicators

Changed to

*Rockglaciers are landforms that form as a result of creeping mountain permafrost which have received considerable attention concerning their dynamical and thermal changes.* |
| M1.4 | Page: 2 **Insert:** (rock wall retreat)

Changed |
| M1.5 | Page: 2  Highlight: crevasse Text: I understand that others have used the term crevasse for rock glaciers, but I suggest to use tension cracks, similar to what is used in landslide literature. Since you are using rockglacier to clearly differentiate the fact that a rock glacier is not a special glacier Io think it would also be appropriate to use a from the landslide / mass movement community to describe the cracks.

Changed |
| M1.6 | Page: 2 Text: what about effect of water.

Changed to

*These potential factors are most likely connected to the complex combination of the local topography, the thermal state of the permafrost (climate-induced response), the existence and intrusion of liquid water and/or to variations in the sedimentation regime affecting the sediment load during long-term landform evolution.* |
| M1.7 | Page: 2  Highlight: constraint Text: really "constraints"

Changed to

*foundation* |
| M1.8 | Page: 3 Delete: coarse

Changed |
| M1.9 | Page: 3 Delete: permafrost

Changed |

| | |
|---|---|
| M1.10 | Page: 3 Delete: ing |
| M1.11 | Page: 3 Text: and presence of moisture

Changed |
| M1.12 | Page: 4 Highlight: rockglacier Huhh1 in the Turtmann valley and the well studied Murtèl rockglacier in the Engadine Text: change order so that it is similar to the order you have in the paper

Changed |
| M1.13 | Page: 4 Text: WGS84?

Changed |
| M1.14 | Page: 4 Insert:

Changed |
| M1.15 | Page: 4 Highlight: mm Text: add space between different units, please check the whole document.

changed |
| M1.16 | Page: 4 Text: Arenson et al., 2002

changed |
| M1.17 | Page: 4 Highlight: l Text: please check the format of all your citations, e.g. et al., 20--

changed |
| M1.18 | Page: 4 Delete: a

changed |
| M1.19 | Page: 4 Insert: an

changed |
| M1.20 | Page: 5 Text: projection?

changed |
| M1.21 | Page: 6 Highlight: a negative subsidence or settlement would actually be heave. I recommend to |

| | not use negative values if the noun you are using has a direction. it would be different if you were to use vertical deformations only and define positive deformations as upward. |
|---|---|
| | changed |
| M1.22 | Page: 8 Text: and temperatures the warmest |
| | changed |
| M1.23 | Page: 9 Text: What is the difference between the surface flow speed and the averaged horizontal flow speed? would the latter be the average of all the surface flow speeds? |
| | They are not the same, one is the velocity at the surface and the other is actually the depth or vertically averaged velocity. For making this clearer we changed this to: |
| | …depth averaged horizontal flow … |
| M1.24 | Page: 9  Highlight: Text: roh |
| | changed |
| M1.25 | Page: 9 Text: You may also indicate that in a glacier A is only temperature dependent (Arrhenius equation) with B being constant, whereas in a rock glacier this parameter may also vary with depth in response to changing ice contents |
| | We agree with the reviewer but explained the varying shearing with depth further above. However, we added a sentences on the point of the reviewer, the text reads now as: |
| | Although this equation is in its form identical to the case of glacier ice (Cuffey and Paterson, 2010), for rock glaciers the flow exponent n and the rate factor A (referring to the material softness) are, due to the presence of debris and water within the ice, not necessarily the same (Moore 2016) and may in reality also vary with depth. |
| M1.26 | Page: 10  Highlight: 50m Text: space between number and unit |
| | changed |
| M1.27 | Page: 13 Highlight: over what time? |
| | Within the next 50 a |
| | See |
| | Marmy, A., Rajczak, J., Delaloye, R., Hilbich, C., Hoelzle, M., Kotlarski, S., Lambiel, C., Noetzli, J., Phillips, M., Salzmann, N., Staub, B., and Hauck, C.: Semi-automated calibration method for modelling of mountain permafrost evolution in Switzerland, The Cryosphere Discuss., 9, 4787–4843, doi:10.5194/tcd-9-4787-2015, 2015. |

| M1.28 | Page: 13 Delete: °

changed |
|---|---|
| M1.29 | Page: 13 Insert: ,

changed |
| M1.30 | Page: 13 Delete: two roughly

changed |
| M1.31 | Page: 13 Insert: about two

changed |
| M1.32 | Page: 17 Text: add Arenson et al., 2002

changed |
| M1.33 | Page: 18 Delete: of

changed |
| M1.34 | Page: 18 Text: how are these changes in relative terms, i.e. percentage? are the relative changes similar for the two rock glaciers?

Yes the relative changes are similar between rock glaciers which we already state just in the sentence above. We do not give relative % numbers because for different quantities they are not the same, but it can be well seen in fig 10 which is already referenced. We rephrased the sentences here slightly to make this point clearer:

*Relative to the initial quantities however (pre-perturbation velocity, thickness or advance rates), the dynamic changes are for both rockglaciers very similar (Fig. 10). This means that we should expect dynamic changes of rockglaciers to be scalable by their geometric (thickness) and kinematic characteristics (flow speed).* |
| M1.35 | Page: 18 Insert: thermal

changed |
| M1.36 | Page: 19 Highlight: this is speculative w/o any calculation. Based on my experience it is likely more in the order of daces. Ground temperatures change slowly.

Marmy et al. show that general permafrost degradation with thawing at 10 m, even partially reaching 20 m depths until the end of the century, but with different timing among the sites.

So its rather decades than centuries! |

| | |
|---|---|
| | *Marmy, A., Rajczak, J., Delaloye, R., Hilbich, C., Hoelzle, M., Kotlarski, S., Lambiel, C., Noetzli, J., Phillips, M., Salzmann, N., Staub, B., and Hauck, C.: Semi-automated calibration method for modelling of mountain permafrost evolution in Switzerland, The Cryosphere Discuss., 9, 4787–4843, doi:10.5194/tcd-9-4787-2015, 2015.* |
| M1.37 | Page: 19 Highlight: to centuries

See M1.36 but the thickness of Murtel might still result in longer warming time scales. |
| M1.38 | Page: 19 Highlight: unclear, check wording

Rewritten to:

*The modeling shows that the adjustment times in thickness and advance rates are for the faster and steeper Huhh1 rockglacier with 100 a, an order of magnitude faster than the 1000 a of Murtèl rockglacier (Fig. 6d and 7d).* |
| M1.39 | Page: 22 Highlight: instead Text: how about: "in addition to" so that these monitoring systems complement each other?

changed |
| M1.40 | Page: 22 Text: The model has one key limitation that must be discussed here, which is that it cannot generate tension cracks. You can only model a continuum and the tension cracks that you are describing at the beginning and that may change rock glacier morphology cannot be reproduced. I do not believe that takes anything away from your conclusions, but it should be explained.

Yes we agree with the reviewer, and clarified this point by adding a sentence in the conclusions that also picks up the issue of internal ice melt from reviewer 2. Otherwise we made it clear that we cannot reproduce individual and small scale features in the model description.

*Note that effects of internal ice melt or non-continuous deformations such as the formation of tension cracks are not included in the current version of the model.* |
| M1.41 | Page: 22 Highlight: alpin Text: can be deleted, as it may also be applicable for rock glaciers in other regions

Changed |
| M1.42 | Page: 22 Highlight: immediately Text: you may have to be careful in saying "immediately". In your model, you are shocking the system, whereas in reality, ground temperatures do not chance so quickly, in particular at depth where the majority of the deformation seem to occur

We agree with the reviewer. It is still feasible to assume that a warming is influences the entire landform faster than a change in material input. Nevertheless, we still deleted the 'immediately' |

| M1.43 | Page: 22  Highlight: sustenance Text: or sustainability?

Changed to endurance |
|---|---|
| M1.44 | Page: 28 **Insert:** "data from sources cited in the text" or did you measure these values by yourself?

We did measure most of the parameters ourselves. But ages and Rockwall properties are published elsewhere for Murtel.

*The data sources are cited in the text.* |
| M1.45 | Page: 31 Highlight: use subscript

changed |
| M1.46 | Page: 32 Text: can you make the elevation change layer transparent so that one can see through and identify some of the structures?

If we make the elevation change more transparent you don't see any of the small scale features. For a more detailed look at the landforms we included the same figure without the elevation change in the supplement Figure S1

Figure S1: Orthophotos oft he two selected rockglaciers. |
| M1.47 | Page: 34 Text: the axis implies that the rock glacier has a single temperature, but this is not |

| | |
|---|---|
| | correct and the axis is actually referring to the temperature of the frozen soil, i.e. the rock glacier material.

Changed |
| M1.48 | Page: 35 Text: why is the bedrock above the DEM surface?

See R2.29 |
| M1.49 | Page: 36 Text: Murtèl, be sure you use correct spelling all the time.

Changed |
| M1.50 | Page: 37 Text: for these plots, would there be value in adding change in velocity as well?

We actually looked at this but it does not add that much and in our view would give too much repetition. We also think reading the absolute velocities is easier and provides the general flow speed of the rock glacier as well. So we did not add such figures. |
| M1.51 | Page: 40 Text: maybe add a note that the change in temperature is immediate, i.e. the system is shocked.

We clarified this in the caption:

*… immediately after a 1°C temperature step increase* |
| | |

**3    Response to Comments in Manuscript C. Kinnard (Reviewer 2)**

| | |
|---|---|
| M2.1 | Page 1 rockglacier or rock glacier? check correct terminology
See R2.1 |
| M2.2 | Page 4 fix coordinates. If in UTM projection then it should be in meters. If in lat/long then use the degree symbol

Changed to

*401 555, 5 115 642 (zone 32T) for Huhh1 and 563 131, 5 142 001 (zone 32T) for Murtel* |
| M2.3 | that is rather dense :)
g m-3
But I think the S.I. is indeed kg m^-3 so values should read 2650-2750 kg m^-3 |

| | |
|---|---|
| | Corrected |
| M2.4 | are theses ages B.P.?

5000 to 6000 years

Indeed B.P. Corrected |
| M2.5 | Page 5: The vertical elevation change is obtained by subtracting the surface-parallel component of the vertical displacement from the total measured vertical displacement. |
| M2.6 | Page 6: I don't see why the ridge and furrow topography is a problem if your are concerned about average topographic changes. Clearly, from your Fig.2 there is advection of topographic features which cause alternating zones of subsidence and rising of the surface, but these will cancel out when averaging. I think, since changes are small, that differences between DEM differencing and the points survey is a results of differences in spatial sampling (density and pattern), i.e. 20 points for fields surveys VS XX gridpoints in DEM differencing. You would need a complete error assessment to be able to truly compare these two measures (measurement error + sampling error). The extraction of vertical changes from the 3D measurements can also be affected by the slope calculations which you used to remove the surface-parallel component of vertical movement.

It would be useful to expand on the reasons why these two dataset give different results, even if you don't do a complete error analysis.

See R2.4 |
| M2.7 | Page 8: why the change in symbology? This is the 'driving stress.? check for consistency

We rephrased it consistently with the shear stress before integrating it. It now reads as:

*The shear stress $\tau_{xz}$ is then given by the local surface slope $\frac{\partial s}{\partial x}$ and ice depth $d_z$*

$$\tau_{xz} = \rho_r g \frac{\partial s}{\partial x} d_z \qquad\qquad (3)$$

*where $\frac{\partial s}{\partial x}$ is the surface slope.* |
| M2.8 | Page 8: here you also make the assumption that density is constant over depth. You should state it clearly.

We state this now more clearly at top of page 8:

*…rockglacier material to be a homogenous mixture of ice and sediment, meaning the rheological parameters such as the rate factor $A$, flow exponent $n$ and rock glacier material density $\rho_r$ do not change within the rockglacier body.* |
| M2.9 | Page 9 I strongly suggest your consult Monnier and Kinnard (accepted 2016) and Moore (2014) for an expanded discussion of the effect of ice, debris and water content on the rheology of ice-debrismixtures.

Indeed with •>60% ice inter-particle frictional strength will not dominate the strength of the debris-ice mixture and the mixture will mostly behave as ice according to Glen's law. The

'softness' parameter (A) will the depend on ice temperature water content, fabric, etc.

Monnier and Kinnard, accepted 2016: Interrogating the time and processes of development |

| | of the Las Liebres rock glacier, central Chilean Andes, using a numerical flow model. Earth Surface processes and Landforms |
|---|---|
| | Moore PL. 2014. Deformation of debris–ice mixtures. Review of Geophysics 52: 435-467 |
| | *See general response to R2.5: we now mention the effect of moisture on deformation and added the two references* |
| M2.10 | Page 9: and other factors, see Hooke (2005) or Cuffey and Paterson (2010) for a discussion for ice, and Monnier and Kinnard for the influence of moisture and its interactions with the debris-ice fraction. |
| | *See general response to R2.5: we now mention the effect of moisture on deformation and added the two references:* |
| M2.11 | Page 10: can it be negative due to melting of internal ice? |
| | *That is correct but we assume the state of an 'intact' rockglacier which is constantly receiving sediment and ice and therefore has a positive accumulation in the deposition zone. Negative accumulation would result subsidence in the upper part of the rockglacier which we interpret as degradation and address later in the .* |
| M2.12 | Page 10: Here you should discuss the implication of side drag, because your SIA formulation assumes an infinite width yet you apply it to a finite width of size w and having parabolic cross-profile. |
| | One manner to deal with this is to include of side drag effect using a shape factor (Monnier and Kinnard 2016 and references cited in their paper) |
| | *This has been addressed, see response R2.7 above.* |
| M2.13 | Page 10: could you not have melting and runoff of internal ice? |
| | *We explicitly excluded internal ice melt and clarified this in the model description (see response . Yes in nature this could be possible which is also what we conclude at the end.* |
| | *We therefore added at the of the discussion section 6.2.4 (geometry change and subsidence)* |
| | *…and therefore we did not include this process in our flow modelling. In future work, ice melt (as a function of temperature) could technically easily included through a negative accumulation term in the surface evolution equation (8) and an adjustment of the mean density.* |
| M2.14 | Page 10: Or gained (?) |
| | *Correct. Changed to* |
| | *Is lost or gained* |
| M2.15 | Page 11: Here you should mention that is stays constant in space and time |
| | *Changed to* |
| | *We estimate the density of the rockglacier material $\rho_r$ from the percentage ice content $c_i$ and from the respective densities of ice $\rho_i = 910$ kg m$^{-3}$ and the debris material $\rho_d = 2700$ kg m$^{-3}$ which we assume to stay constant in space and time* |

| M2.16 | Page 11: also see Monnier and Kinnard and Moore (2014) : increasing debris concentration can increase moisture content, which decreases the viscosity, up to the limiting debris fraction fromwhich frictional strength increase. |
| | Has been addressed, see response R2.5 |
| M2.17 | Page 12: Is this explained somewhere? Your flow equation does not include threshold for thickness, is the 'critical' thickness only a results of the nonlinear flow law? Please explain better. |
| | There is not a fixed critical thickness but the transition is rather smooth. We therefore changed it to: |
| | *... once it is thick enough and the shear stress high enough,...* |
| M2.18 | Page 13: You don't take into account that as the rock glacier expands downslope/down valley temperatures are warmer there which will feedback on internal ice melting and rheology. Your figure 6 suggest that under a constant accumulation rate the rock glacier keeps increasing in length over time. Is this realistic? Would you not expect increased melting of internal ice as the front expands downslope, and a parallel increase in frictional strength? Theses aspects should at least presented in the discussion |
| | Yes, as we exclude melting of ice in our modelling experiments, the rock glacier would keep advancing. For the time scales and related rock glacier length considered here the elevation at the tongue does however not change significantly and therefore unlikely affects melt rates. We added a sentence in section 5.1 (Rock glacier build up) clarifying thjs point: |
| | *This continuing advance is a direct result of constantly adding mass at the top while no mass is remobved through melting.* |
| M2.19 | Page 15: Why did you choose to present this particular scenario? Is it because it is the one that best fit the observations? Is there any theoretical and observational evidence for decreased material input? Is climate warming not causing increased rockfall in the Alps? My impression is that the rockfall rate would increase, at least initially, but that the ice content (%) would decreases if snowfall rates decrease as climate warms... Hence the assumption off a constant ice content for the accumulation rate, ac, may be simplistic. That being said it is ok to simplify as long as these simplifications and their potential effects are presented and discussed. |
| | See R2.8 |
| M2.20 | Page 17: but see recent review by Moore (2014) |
| | Moore (2014) gives a very comprehensive overview the physical properties and mechanical interactions between ice and rock debris. His summary concerning rockglaciers and their special characteristics relies on almost the same studies as we used in our experiment (such as Arenson, 2002, Arenson & Springman, 2005 and Ikeda ). |
| | He interestingly stresses the point of the role of unfrozen water influencing the mechanical properties of rock-ice mélanges and presents water concentration of several frozen homogeneous materials and elaborates on its impact on rheology. This is surely the right direction for further research but his recommendations are hardly applicable to our rockglacier rheology since the main source of unfrozen water in rockglacier is precipitation and snow melt and there we is to our knowledge no field based data yet how rain- or melt-fed subsurface hydrology develops in rockglaciers. |
| | In our further research on RG rheology we will definitely look more detailed into this paper but we still believe that 'there is very limited quantitative, field-based information available |

| | |
|---|---|
| | *to constrain more complex constitutive relationships' concerning the description of the flow law. To elaborate on that we changed the following in the text:*

*A more complex rheology of ice-debris mixtures could in theory and should in the future be included in rockglacier creep models, but currently there is very limited quantitative, field-based information available to constrain more complex constitutive relationships. Especially the role of unfrozen water appears to have a strong impact on the properties ice-debris mixtures as temperature nears the melting point and need to be further addressed (Moore, 2014) in establishing an adapted flow-law for rockglaciers.*

*Moore, P. L.: Deformation of debris-ice mixtures, Rev. Geophys., 52, 435–467, doi:10.1002/2014RG000453, 2014.* |
| M2.21 | Page 18: very different for the two rock glaciers

Change accordingly |
| M2.22 | Page 18: the sentences does not read well I suggest rephrasing

This sentence has been rephrased to:

*The modeling shows that the adjustment times in thickness and advance rates are for the faster and steeper Huhh1 rockglacier with 100 a, an order of magnitude faster than the 1000 a of Murtèl rockglacier (Fig. 6d and 7d).* |
| M2.23 | Page 19: Could you explain better the difference between these two quantities?

Changed the sentence to:

*…which results in 3000 a and 1200 a for Murtèl and Huhh1, respectively, for the thickness perturbation to spread over the entire landform and which is substantially longer than the timescales derived above for the kinematic wave propagation.* |
| M2.24 | Page 20: Why? explain.

We changed this sentence for clarifying:

*These diffusion timescales are also much longer compared to pure-ice glaciers and are consistent with the persistent occurrence of morphological features of ridges and furrows on the surface of rockglaciers.* |
| M2.25 | Page 20: I am confused.

Would the kinematic adjustment time, Tk, not be rather similar to the reaction time for glaciers, i.e. the time it takes for the geometry to respond to a change in mass flux, while the diffusion time, Td, the time taken for the perturbation to spread over the whole landform, i.e. the volumetric response time, which is longer than the reaction time?

I may get this wrong but I suggest you clarify and differentiate these two concepts clearly in the text for the reader

Has been addressed, see response R2.9 |
| M2.26 | Page 21: The DEM differencing over the WHOLE rock glacier should give you the variation in mass no? (geodetic balance). This assuming that they are precise enough. By integrating topographic changes over the whole RG area mass transfers cancel out and the volume change will be the mass-balance, which for a rock glacier is m = material (rock+ice) accumulation - internal ice melting. If m is negative then internal ice melting was more than the accumulation rate. If m is positive you can compared it with the theoretical value you |

| | |
|---|---|
| | used for accumulation... but of course you cannot isolate internal melting.

See 2.4 |
| M2.27 | Page 22: rephrase.

Do you mean: rock glacier show spatially contrasted response to changes in environmental factors.

We mean that the rockglacier adapts to changes in environmental factors in different ways at different places. Not contrasting but temporally and spatially variable.

*Rockglaciers react spatially and temporally variable to changes in environmental factors.* |
| M2.28 | Page 22: Consult Moore (2014) for a more up to date review of debris-ice mixture rheology models, and Monnier and Kinnard (2016) for an application. One intervening factor ignored in your analysis is the moisture content. There is not necessarily an easy way to include it, but this aspect should be mentioned and discussed briefly, see Monnier and Kinnard for a discussion of the topic and related key references

We addressed this issue in the main text (R2.5) but do not want to bring this up in the conclusions. |
| M2.29 | Page 23: the modelled front does not connect with bedrock, why? Does the modelled rock glacier extend further downslope?

For the modelling application we introduced strong limitations according the bedrock topography which are responsible for the deviation between observed and modeled rockglacier topographies. We simplified the talus slope and rockglacier bedrock topography as planes with two constant inclinations. The talus slope bedrock inclined with the angle of repose for such material as stated in Carson, 1977 and the rockglacier bedrock derived from field-based geomorphometry at the two sites. The real bedrock topography is most likely more complex than this assumption which results in the deviating model results. In particular the lower part of Murtel rockglacier is believed to have a less inclined bedrock topography and even convex formation at the snout of the rockglacier which is hindering its advance. In the case of the talus slope of Huhh1 where the observed topography lies lower than the modeled surface, we introduced an idealized upper talus slope (first 150 m) in the modelling which deviates from the observed surface. We believe the upper part of the rockglacier to be already disconnected from its source area, therefore 'draining' the talus slope of material and the rockglacier body 'creeping away' which would explain the deviation between model and observation.

The point we are trying to make with this figure is that the model produces a good agreement concerning length and thickness in the assumed timespan of rockglacier development.

In order to clarify we added the following in the text:

*Note that the bedrock topography of the talus slope and rockglacier are assumed to be of constant inclination. We defined the first 150 m of slope as an idealized talus slope with an inclination of 37° which we assumed to be the angle of repose for such unsorted and unconsolidated blocky material (Carson, 1977). The general inclination of the bedrock topographies (see Table 1) underneath the rockglaciers are derived from digital elevation models where we assume the bedrock to be parallel to the rockglacier surface (see Table 1). Figure 5 depicts the observed rockglacier front shapes and positions slightly differently from the modeled ones as the real bedrock geometries of the rockglaciers are more complex than* |

| | |
|---|---|
| | *the assumed uniform mountain slopes. Also, the idealized talus slopes differ from the observed geometries due to above mentioned simplification.*

And in the caption of fig 5:

*Note that the model cannot replicate the exact geomoetry of the landform due to tis various simplifications but shows good agreement in length and thickness of the rockglacier main body.* |
| M2.30 | Page 23: why is the bedrock surface above the DEM?

See M2.29 |
| M2.31 | Page 36: the assumption of mass conservation ignores spatial variations in internal ice melting rates. The active-layer thickness is likely to increase downslope as air temperatures become warmer: this will reduce the overall rock glacier density (if ice is replaced by air...) and eventually increase the frictional strength as the ice content decreases.

Addressed, see response in R2.6 |
| M2.32 | Page 37: to be consistent with Figure 6 you should move panel d (length) to the left and panel e (volume) to the right

Changed |

---

## Author Response (AR2)

**Author response to reviewer comments, round 2**

In the following we respond to the reviewers comments on our manuscript. L. Arenson's comments were more technical whereas C. Kinnards comments needed some elaboration. Thank you very much for the attentive revision and we hope we incorporated the comments satisfactorily.

Note for notation in our response below: In blue our response to the reviewers comments and italics marks the text as it has been changed to in the revised paper.

**1   Response to Comments Review 2 L. Arenson (Referee 1)**

| R1.1 | - Page 13, line 2: "below" instead of "underneath"

 changed |
|------|---------------------------------------------------------------------|
| R1.2 | - Page 15, line 19: "Twelve" instead of "12". Note: Don't start a sentence with a numeric value.
 changed |
| R1.3 | - Page 18, line 8 & 9. Please check and reword this new sentence.
 Changed to: *The flow-law exponent n has been found to increase with ice content (Arenson et al., 2002, Arenson and Springman, 2005) and rockglacier creep is mostly dominated by relatively thin shear layers with reduced viscosity (Hoelzle et al., 2002, Haeberli et al., 2006, Buchli et al., 2013)* |
| R1.4 | - Whole document: "Timescale", "time scale" or "time-scale"?
 changed |
| R1.5 | - Whole document: "adjustment times", " 'adjustment time-scale' " or " 'adjustment times' "?
 changed to adjustment timescale |
| R1.6 | - Page 23, line 4: "longevity"or "permanence" instead of "endurance"
 changed to longevity |
| R1.6 | - Caption Figure 5: "geometry" instead of "geomoetry"
 changed |
| R1.7 | - Figure 6 to 8 & 10 to 11: In the Figure use the proper Murtèl spelling with the accent.
 changed |

| R1.8 | - Figure 6e: x-Axis label is cut |
| | changed |

| R1.9 | - Caption Figure 8: Space between number and unit |
| | changed |

| R1.10 | - Caption Figure 8: "introduced. The lines" instead of "introduced The lines" |
| | changed |

| R1.11 | - Figure 10: There is a grey box that should be deleted. |
| | Kind regards and good luck with further research. |

**2 Response to Comments in Author Response C. Kinnard (Referee 2)**

I attached a few comments to the authors regarding some points that were adressed only partially in my opinion (effect of ignoring side drag on 'A' parameters and related sensitivity experiments).

The discussion of the influence of debris/ice fraction on the rheology and hence temperature sensitivity of rock glaciers should be discussed with a bit more details than just a couple of reference, as was suggested by both reviewers. The impact of moisture variations as the ice/debris fraction changes could have impacts on the rheology not considered in this model. I do not expect the authors to change their results and still consider their study very relevant, but the authors could strenghten the discussion about their 'model limitations and implications for rock glacier temperature sensitivity'...

We added the following paragraph to the manuscript elaborating on the role of moisture and water. We also mention that we neglect the role unfrozen water/moisture and its impacts in the section 'model setup'. Therefore we think that a comprehensive discussion of its role is more confusing than helpful for the purpose of our simple model approach.

*This becomes increasingly more complex as an expected warming will not only influence the rheological properties of the ice itself but also change the ratio of ice and debris by reducing the volumetric ice content. A new ice/debris proportion will alter the viscosity of the rockglacier in a spatially heterogeneous manner because melting effects have been shown to be spatially diverse (Arenson and Springman, 2005 and Monnier and Kinnard, 2016). The direct impact of moisture variations due to precipitation events has been shown to impact rockglacier rheology on a short term down to a few days (Wirz et al., 2015), whereas the impact of a changing ice/water proportion is assumed to show its consequences on the long-term.*

*Nevertheless, due to the fact that the creep is dominated near the base within our model and that we have calibrated our model parameters to observed geometry and velocities, we do not expect the general dynamical behaviour and involved timescales to be substantially different for other rheological parameter choices.*

We also implemented the following references:

Monnier, S. and Kinnard, C.: Interrogating the time and processes of development of the Las Liebres rock glacier, central Chilean Andes, using a numerical flow model, Earth Surf. Process. Landforms, 41, 1884–1893, doi:10.1002/esp.3956, 2016.

Wirz, V., Gruber, S., Purves, R. S., Beutel, J., Gärtner-Roer, I., Gubler, S., and Vieli, A.: Short-term velocity variations of three rock glaciers and their relationship with meteorological conditions, Earth Surf. Dynam. Discuss., 3, 459–514, doi:10.5194/esurfd-3-459-2015, 2015.

Arenson, L. U. and Springman, S. M.: Mathematical descriptions for the behaviour of ice-rich frozen soils at temperatures close to 0 °C, Canadian Geotechnical Journal, 42, 431–442, doi:10.1139/t04-109, 2005.

| | |
|---|---|
| R2.1 | Page 2: is this -0.02? Check final manuscript

Changed to -0.02 |
| R2.2 | Page 3: -0.02?

Changed |
| R2.3 | Page 5: As far as I see you only derived A from inversion of equation 6 and did not calculate f_A (results in table 3). This is actually confusing in the manuscript: you introduce f_A in equation 6 but is does not re-appear afterward?

I disagree that ignoring side drag would have no effect. Since its effect is presently captured in the A parameter, ignoring side drag gives you lower A values than if you introduce a shape parameter. This is because side drag lowers the total stress from the rock glacier column, and hence your inverted A values must be lower. Lower A values will impact you sensitivity experiments...

We agree that explicitly including side drag (by introducing a form factor) would change the inverted A value specifically the scaling factor $\cdot f_A$. Nevertheless, such a form factor would only act as a scaling factor, that is kept constant over time, and which is already implicitly included in our A. Therefore,  all the modelling results and sensitivities would be exactly the |

| | | |
|---|---|---|
| | same. | |
| R2.4 | Page 5: ... which is calibrated on surface displacements.

changed | |
| R2.5 | Page 6: attenuate..?

changed | |
| R2.6 | Page 6: reduce the ice contribution (or fraction) to the total material influx.

Changed to: *Therefore we keep the material input from the rockwall constant but reduce the ice contribution to the total material influx.* | |